# Causal Information Prioritization for Efficient Reinforcement Learning

**Hongye Cao**[1]  **Fan Feng**[2,3]  **Tianpei Yang**[1,4*]  **Jing Huo**[1]  **Yang Gao**[1,4]

[1]National Key Laboratory for Novel Software Technology, Nanjing University
[2]University of California, San Diego   [3]MBZUAI
[4]School of Intelligence Science and Technology, Nanjing University

## Abstract

Reinforcement Learning (RL) methods often suffer from sample inefficiency, one of the underlying reasons is that blind exploration strategies may neglect causal relationships among states, actions, and rewards. Although recent causal approaches aim to address this problem, they lack grounded modeling of reward-guided causal understanding of states and actions for goal orientation, thus impairing learning efficiency. To tackle this issue, we propose a novel method named Causal Information Prioritization (`CIP`) that improves sample efficiency by leveraging factored MDPs to infer causal relationships between different dimensions of states and actions with respect to rewards, enabling the prioritization of causal information. Specifically, `CIP` identifies and leverages causal relationships between states and rewards to execute counterfactual data augmentation to prioritize high-impact state features under the causal understanding of the environments. Moreover, `CIP` integrates a causality-aware empowerment learning objective, which significantly enhances the agent's execution of reward-guided actions for more efficient exploration in complex environments. To fully assess the effectiveness of `CIP`, we conduct extensive experiments across 39 tasks in 5 diverse continuous control environments, encompassing both locomotion and manipulation skills learning with pixel-based and sparse reward settings. Experimental results demonstrate that `CIP` consistently outperforms existing RL methods across a wide range of scenarios. The project page is https://sites.google.com/view/rl-cip/.

## 1 Introduction

Reinforcement Learning (RL) has emerged as a powerful paradigm for training intelligent decision-making agents to learn optimal behaviors by interacting with their environments, receiving reward feedback, and iteratively optimizing their decision-making policies (Haarnoja et al., 2018; Sutton, 2018; Silver et al., 2017; Cao et al., 2023; Hao et al., 2023; Cao et al., 2022). Despite its notable successes, most RL approaches are faced with the sample-inefficiency problem, which means they typically necessitate an enormous number of interactions with the environment to learn policies, which can be impractical or costly in real-world scenarios (Savva et al., 2019; Kroemer et al., 2021). Inefficient policy learning often results from blind exploration strategies that neglect causal relationships, leading to spurious correlations and suboptimal solutions with high exploration costs (Zeng et al., 2023; Liu et al., 2024).

Causal reasoning captures essential information by analyzing causal relationships between different factors, filtering out irrelevant information, and avoiding interference from spurious correlations (Wang et al., 2022; Pitis et al., 2022; Zhang et al., 2024; Huang et al., 2022b). These approaches build internal causal structural models, enabling agents to strategically focus their exploration on the most pertinent aspects of the environment. They significantly reduce the number of samples required and demonstrate remarkable performance in single-task learning, generalization, and counterfactual reasoning (Richens & Everitt, 2024; Urpí et al., 2024; Deng et al., 2023; Huang et al., 2022a; Feng & Magliacane, 2023). However, most of these works overlook the reward-relevant causal relationships among different factors, or only partially consider the causal connections between states, actions, and rewards (Liu et al., 2024; Ji et al., 2024a), thus hindering efficient exploration.

---

*Corresponding to Tianpei Yang (`tianpei.yang@nju.edu.cn`).

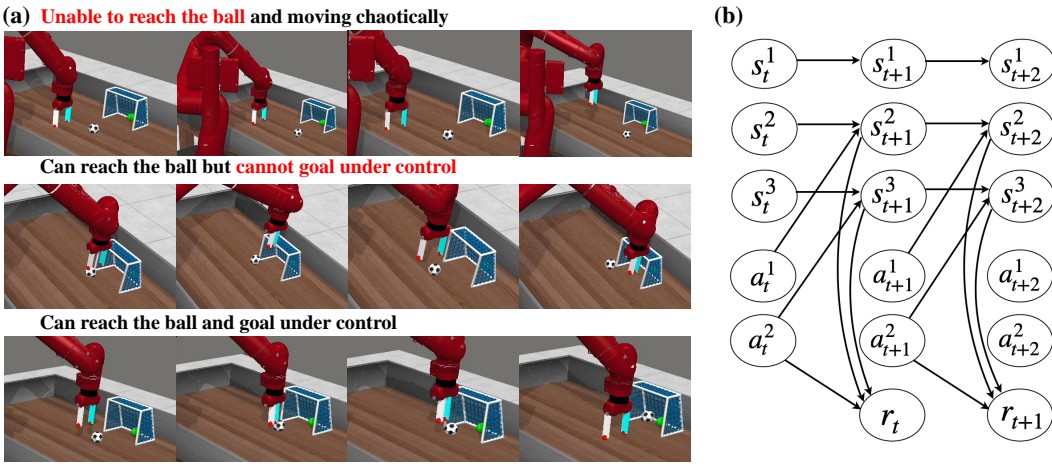

Figure 1: (a). An example of a robot manipulation soccer task with three trajectories, where the objective is to move the ball into the goal. (b). Underlying causal structure of this example in a factored MDP. Different nodes represent different dimensional states and actions.

In this work, we aim to identify and exploit task-specific causal relationships between states, actions, and rewards, enabling agents to discern relevant states and select actions that maximize rewards, ultimately facilitating precise and goal-oriented behaviors. Here we provide a motivating example in Figure 1, showing three trajectories for executing a manipulation soccer task, along with the underlying causal structure in a factored Markov Decision Process (MDP) (Kearns & Koller, 1999). In the first trajectory (row 1), when the agent fails to distinguish states with more intricate causal relationships of the task, the robotic arm exhibits chaotic moving and receives no rewards. The second trajectory (row 2) shows that even without chaotic movements, uncontrollable actions unrelated to the reward lead to an inability to guide the ball towards the goal. Only by filtering out irrelevant state features and executing more controllable actions can we guarantee that the ball is kicked into the goal like row 3. Quantifying the contribution of different factors to the reward can effectively help analyzing important causal relationships.

To address the limitation of sample-inefficiency and leverage the potential of causal reasoning, we propose a novel approach named Causal Information Prioritization (**CIP**) for efficient RL, improving learning efficiency from the perspective of rewards. Building upon the factored MDPs, **CIP** infers causal relationships between states, actions, and rewards across different dimensions, respectively. **CIP** employs counterfactual data augmentation based on the causality between states and rewards to generate transitions, prioritizing critical state transitions. Furthermore, **CIP** leverages the causality between actions and rewards to reweight actions, while utilizing empowerment to maximize mutual information between causally informed actions and future states, thereby enabling better control.

Specifically, **CIP** leverages collected data to construct a reward-guided structural model that explicitly reasons about state-reward causal influences, enabling the swapping of causally independent state features across observed trajectories to generate synthetic transitions without additional environment interactions. By swapping independent state features across different transitions (i.e., irrelevant state dimensions of chaotic movements in the soccer task), **CIP** accentuates causally dependent state information (i.e., relevant states to reach the ball), facilitating focused learning of critical state transitions. Subsequently, **CIP** constructs another structural model that incorporates actions and rewards to reweight actions of dimensions. To enhance the exploration efficiency, **CIP** integrates a causality-aware empowerment term, quantifying the agent's capacity to exert controlled influence over its environment through the mutual information. This empowerment term, combined with causally weighted actions, is integrated into the learning objective, prioritizing actions with high causal influence. The synthesis of causal reasoning and action empowerment enables agents to focus on behaviors that are causally relevant to the task, leading to more efficient and effective policy learning. The main contributions of this work can be summarized as follows.

- To address limitations of blind exploration and sample-inefficiency, we introduce **CIP**, a novel efficient RL framework that prioritizes causal information through the lens of reward. **CIP** bridges the gap between causal reasoning and empowerment to facilitate efficient exploration.

- **CIP** constructs reward-guided structural models to uncover causal relationships between states, actions, and rewards across dimensions. By leveraging state-reward causality, it performs counterfactual data augmentation, eliminating the need for additional environment interactions, and enabling learning on critical state transitions. Exploiting action-reward causality, it reweights actions to enhance exploration efficiency through empowerment. By prioritizing causal information, **CIP** enables agents to focus on behaviors that have causally significant effects on their tasks.
- To validate the effectiveness of **CIP**, we conduct extensive experiments in 39 tasks across 5 diverse continuous control environments, including manipulation and locomotion. These comprehensive evaluations demonstrate the effectiveness of **CIP** in pixel-based and sparse reward settings, underscoring its versatility and reliability.

## 2 RELATED WORK

### 2.1 CAUSAL RL

The application of causal reasoning in RL has shown significant potential to improve sample efficiency and generalization by effectively excluding irrelevant environmental factors through causal analysis (Huang et al., 2022a; Feng & Magliacane, 2023; Mutti et al., 2023; Sun et al., 2024; Sun & Wang). Wang (Wang et al., 2021) introduces a novel regularization-based method for causal dynamics learning, which explicitly identifies causal dependencies by regulating the number of variables used to predict each state variable. CDL (Wang et al., 2022) takes an innovative approach by using conditional mutual information to compute causal relationships between different dimensions of states and actions. IFactor (Liu et al., 2024) is a general framework to model four distinct categories of latent state variables, capturing various aspects of information. ACE (Ji et al., 2024a), an off-policy actor-critic method, integrates causality-aware entropy regularization. Table 2 provides a categorization of various causal RL methods, highlighting their focus on different reward-guided causal relationships. Existing approaches do not fully account for the causal relationships between both states and actions with rewards. Our goal is to explore these causal relationships from a reward-guided perspective to enhance sample efficiency across a broader range of tasks.

### 2.2 EMPOWERMENT IN RL

Empowerment, an information theory-based concept of intrinsic motivation, has emerged as a powerful paradigm for enhancing an agent's environmental controllability (Mohamed & Jimenez Rezende, 2015; Klyubin et al., 2005; Cao et al., 2024). This framework conceptualizes actions and future states as information transmission channels, offering a novel perspective on agent-environment interactions. In RL, empowerment has been applied to uncover more controllable associations between states and actions, as well as to develop robust skill (Salge et al., 2014; Bharadhwaj et al., 2022; Choi et al., 2021; Eysenbach et al., 2018; Leibfried et al., 2019; Seitzer et al., 2021). Empowerment, expressed as maximizing mutual information $\max_\pi I$, serves as a learning objective in various RL frameworks, providing intrinsic motivation for exploration and potentially yielding more efficient and generalizable policies. Our approach extends empowerment in RL by examining the influence of state, actions, and rewards through a causal lens, integrating causal understanding with empowerment to enhance exploration strategy and learning efficiency.

### 2.3 OBJECT-CENTRIC RL AND OBJECT-ORIENTED RL

Recent advances in object-centric representation learning focus on acquiring and leveraging structured, object-wise representations from high-dimensional observations. Foundational works include Slot Attention (Locatello et al., 2020) and AIR (Eslami et al., 2016; Kosiorek et al., 2018), establishing basis for this field. Subsequent follow-ups have worked on these concepts by employing state-of-the-art architectures, including DINO-based approaches Zadaianchuk et al. (2023), transformer-based models (Wu et al., 2022), diffusion models (Jiang et al., 2023), and state-space models (Jiang et al., 2024). Notably, learning object-centric representations can enable compositional generalization across various domains, such as video and scene generation (Wu et al., 2023; 2024). Moreover, several theoretical studies have explored the mechanisms underlying compositional generalization and the causal identifiability (Kori et al., 2024; Brady et al., 2023; Lachapelle et al., 2024).

Object-centric representations have been effectively employed in world models to capture multi-object dynamics, as demonstrated by works (Jiang et al., 2019; Lin et al., 2020; Kossen et al., 2019). Building on these object-centric world models, various studies use them in RL by better modeling complex object-centric structures in partially observable MDPs (Kossen et al., 2019; Mambelli et al., 2022; Feng & Magliacane, 2023; Choi et al., 2024), identifying critical objects (Zadaianchuk et al.,

2022; Park et al., 2021), and learning object-centric policies (Zadaianchuk et al., 2021; Yuan et al., 2022) and applications in robotic manipulation tasks (Li et al., 2020; Mitash et al., 2024; Haramati et al., 2023; Li et al., 2024), as well as in learning intrinsic or curiosity-driven policies based on objects and their interactions (Watters et al., 2019; Wang et al., 2024c;b).

Another research direction explores object-oriented MDPs, with the homomorphic object-oriented world model being a notable example that leverages MDP homomorphism to model object dynamics and enable efficient planning through symmetric equivalence in MDPs (Diuk et al., 2008; Scholz et al., 2014; Wandzel et al., 2019; Van der Pol et al., 2020; Rezaei-Shoshtari et al., 2022; Zhao et al., 2022), provides a powerful foundation for learning object-oriented MDPs and facilitates efficient planning (Wolfe & Barto, 2006). Our work, which focuses on uncovering general causal relationships among components in MDPs and empowerment optimization, is orthogonal to object-centric RL. However, object-centric RL could provide useful abstract object-based variables that could be useful for causal structure learning in complex environments [1].

## 3 Preliminaries

### 3.1 Markov Decision Process

In RL, the agent-environment interaction is formalized as an MDP. The standard MDP is defined by the tuple $\mathcal{M} = \langle \mathcal{S}, \mathcal{A}, \mathcal{P}, \mu_0, r, \gamma \rangle$, where $\mathcal{S}$ denotes the state space, $\mathcal{A}$ represents the action space, $\mathcal{P}(s'|s,a)$ is the transition dynamics, $r(s,a)$ is the reward function, and $\mu_0$ is the distribution of the initial state $s_0$. The discount factor $\gamma \in [0, 1)$ is also included. The objective of RL is to learn a policy $\pi : \mathcal{S} \times \mathcal{A} \to [0, 1]$ that maximizes the expected discounted cumulative reward $\eta_{\mathcal{M}}(\pi) := \mathbb{E}_{s_0 \sim \mu_0, s_t \sim \mathcal{P}, a_t \sim \pi} \left[ \sum_{t=0}^{\infty} \gamma^t r(s_t, a_t) \right]$.

### 3.2 Structural Causal Model

A Structural Causal Model (SCM) (Pearl, 2009) is defined by a distribution over random variables, defined as $\mathcal{V} = \{s_t^1, \cdots, s_t^d, a_t^1, \cdots, a_t^n, r_t, s_{t+1}^1, \cdots, s_{t+1}^d\}$ and a Directed Acyclic Graph (DAG) $\mathcal{G} = (\mathcal{V}, \mathcal{E})$ with a conditional distribution $\mathcal{P}(v_i|\text{PA}(v_i))$ for node $v_i \in \mathcal{V}$. Then the distribution can be specified as:

$$p(v_1, \ldots, v_{|\mathcal{V}|}) = \prod_{i=1}^{|\mathcal{V}|} p(v_i|\text{PA}(v_i)), \tag{1}$$

where $\text{PA}(v_i)$ is the set of parents of the node $v_i$ in the graph $\mathcal{G}$.

**Causal Structures in MDP**  We use a factored MDP (Kearns & Koller, 1999; Guestrin et al., 2003; 2001) to model the MDP and the underlying causal structures between states, actions, and rewards. In the factored MDP, nodes represent system variables (rewards and different dimensions of the states and actions), while the edges denote their relationships within the MDP. We employ causal discovery methods to learn the structures of $\mathcal{G}$.

We can identify the graph structure in $\mathcal{G}$, which can be represented as the adjacency matrix $M$. To integrate such relationships in MDP, we explicitly encode the causal mask over variables into the reward function. Hence, the reward function in MDP with the causal structure is defined as follows:

$$r_t = R(M^{s \to r} \odot s_t, M^{a \to r} \odot a_t, \epsilon_{r,t}) \tag{2}$$

where $\odot$ denotes the element-wise product. $M^{s \to r} \in \mathbb{R}^{|s| \times 1}$ and $M^{a \to r} \in \mathbb{R}^{|a| \times 1}$ are the adjacency matrices indicating the influence of current states and actions on the reward, respectively, and $\epsilon_{r,t}$ represents i.i.d. Gaussian noise. Under the Markov condition and faithfulness assumption (Pearl, 2009; Spirtes et al., 2001), the structural vectors are identifiable. The detailed assumptions and propositions can be found in Appendix B. In this work, our objective is to discover and leverage these two causal matrices to prioritize causal information for efficient RL.

### 3.3 Empowerment in RL

Empowerment quantifies an agent's capacity to influence its environment and perceive the consequences of its actions (Klyubin et al., 2005; Bharadhwaj et al., 2022; Jung et al., 2011). In our

---

[1] We provide a detailed discussion in Appendix C.1.

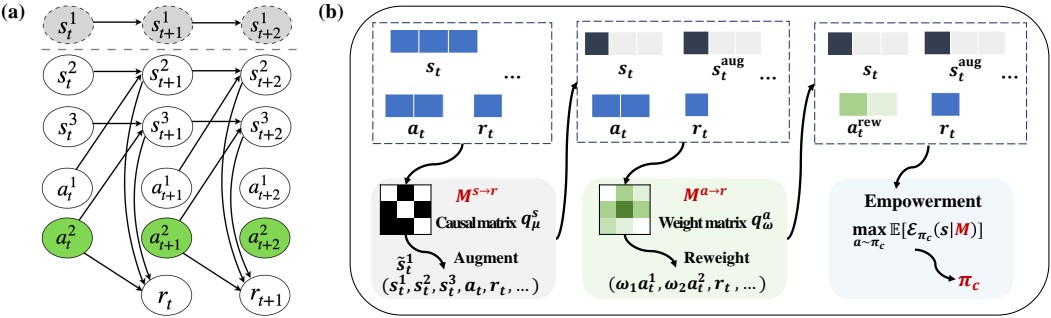

Figure 2: (a) Underlying causal structure of **CIP**. (b) The whole learning process of **CIP** includes counterfactual data augmentation, causal action reweight and causal action empowerment.

framework, the empowerment is defined as the mutual information between the agent state $s_{t+1}$ and action $a_t$, conditioned on the present state $s_t$ and causal mask $M$, as shown follows:

$$\mathcal{E} := \max_{\pi} \mathcal{I}(a_t; s_{t+1} \mid s_t, M), \tag{3}$$

where $\mathcal{E}$ denotes the channel capacity from actions to states. Unlike (Cao et al., 2024), which focuses on action-to-state empowerment effects, we leverage causal understanding and more accurate entropy calculation to analyze state-to-action influences, facilitating the development of more controllable behavioral policies.

## 4 CAUSAL INFORMATION PRIORITIZATION

In this section, we introduce the proposed framework **CIP**, which implements causal information prioritization based on the causal relationships between states, actions, and rewards (as shown in Figure 2). First, we train a structural model based on the causal discovery method, DirectLiNGAM (Shimizu et al., 2011) using collected trajectories to obtain a causal matrix $M^{s \to r}$. Utilizing this matrix, **CIP** executes the swapping of causally independent state features, generating synthetic transitions (Section 4.1). This process of swapping independent state information accentuates causally dependent state information, enabling focused learning on critical state transitions. Subsequently, **CIP** constructs another structural model to get a weight matrix $M^{a \to r}$ that incorporates actions and rewards to reweight actions (Section 4.2). Furthermore, **CIP** integrates a causality-aware empowerment term $\mathcal{E}_{\pi_c}(s)$ combined with causally weighted actions into the learning objective to promote efficient exploration. This integration encourages the agent's policy $\pi_c$ to prioritize actions with high causal influence, thereby enhancing its goal-achievement capabilities.

### 4.1 COUNTERFACTUAL DATA AUGMENTATION

To discover the causal relationships between states and rewards, we initially collect trajectories to train a structural model by the DirectLiNGAM method, denoted as $q_\mu^s$, to obtain the causal matrix $M^{s \to r}$. Subsequently, we infer the local factorization, which is utilized to generate counterfactual transitions. For each state $s$ in the trajectories, we compute the uncontrollable set, defined as the set of variables in $s$ for which the agent has no causal influence on rewards:

$$\mathcal{U}_s = \{s^i \mid M^{s \to r} \cdot (s_t^i, r_t) < \theta; i \in [1, N]\}, \tag{4}$$

where $\theta$ is a fixed threshold and $N$ is the dimension of the state space. The set $\mathcal{U}_s$ encompasses all dimensional state variables for which the causal relationship $s_t^i \to r_t$ does not exist in the causal matrix of states and rewards. Utilizing the learned causal matrix $M^{s \to r}$, we partition all state variables in the factored MDP into controllable and uncontrollable sets. These uncontrollable sets are then leveraged for counterfactual data augmentation, thereby prioritizing the causally-informed state information to improve learning efficiency.

To generate counterfactual samples, we perform a swap of variables that fall under the uncontrollable category (i.e., in set $\mathcal{U}_s$) sampled from the collect trajectories. Specifically, given two transitions $(s_t, a_t, s_{t+1}, r_t)$ and $(\hat{s}_j, \hat{a}_j, \hat{s}_{j+1}, \hat{r}_j)$ sampled from trajectories, which share at least one uncontrollable sub-graph structure (i.e., $\mathcal{U}_s \cap \mathcal{U}_{\hat{s}} \neq \emptyset$), we construct a counterfactual transition $(\tilde{s}_t, \tilde{a}_t, \tilde{s}_{t+1}, \tilde{r}_t)$ by swapping the irrelevant state variables $(s_t^i, s_{t+1}^i)$ with $(\hat{s}_j^i, \hat{s}_{j+1}^i)$ for each $i \in \mathcal{U}_s \cap \mathcal{U}_{\hat{s}}$. The

augmented transitions will be added to the training data for causal reasoning during subsequent action empowerment, thus eliminating the need for additional environment interactions to prioritize causal information. Furthermore, we also consider directly using controllable state sets combined with causal action empowerment to replace counterfactual data augmentation for policy learning. The comparative experimental results validating this approach are presented in Appendix D.3.1.

## 4.2 Causal Action Prioritization Through Empowerment

**Causal action reweight**   Having analyzed the causal relationships between states and rewards to achieve efficient data augmentation, in this section, we further discover the causal relationships between actions and rewards to prioritize causally-informed decision-making behaviors. **CIP** constructs a reward-guided structural model, incorporating states (including augmented states), actions, and rewards. This model forms the foundation for action prioritization in policy learning, enabling action reweighting based on causality. Leveraging this structural model to delineate relationships between policy decisions and rewards, we evaluate the causal impact of different actions on reward outcomes. In this way, the agent focuses on pivotal actions with demonstrable causal links to desired reward outcomes, potentially accelerating learning and optimizing performance in complex environments.

Specifically, in **CIP**, we employ DirectLiNGAM method to train a causal structural model $q_\omega^a$, which yields a weight matrix $M^{a \to r}$, delineating the relationships between actions and rewards, conditioned on the states. For a given set of actions $(a_t^1, a_t^2, a_t^3, \dots)$, we utilize the weight matrix $M^{a \to r}$ to reweight them as $(\omega_1 a_t^1, \omega_2 a_t^2, \omega_3 a_t^3, \dots)$, where $\omega$ represents the causal weights derived from the matrix $M^{a \to r}$. By leveraging this causal structure, we can prioritize the most pivotal actions, potentially leading to more efficient policy exploration and targeted policy improvements.

**Causal action empowerment**   Based on the learned causal structure, we propose the causal action empowerment to incorporate the reweighted actions into the learning objective for efficient exploration in a controllable manner. To this end, we design a causality-aware empowerment term $\mathcal{E}_{\pi_c}(s)$ for policy optimization. We maximize the empowerment gain of the policy $\pi_c$, where $\pi_c$ incorporates the learned causal structure. This approach allows us to quantify and maximize the empowerment that can be achieved by explicitly considering causal relationships, thereby bridging the gap between causal reasoning and empowerment.

We denote the empowerment of the causal policy as $\mathcal{E}_{\pi_c}(s) = \max_a \mathcal{I}(a_t; s_{t+1} \mid s_t; M)$. We then formulate the following objective empowerment function:

$$
\begin{aligned}
\mathcal{E}_{\pi_c}(s) &= \max_a \mathcal{I}(a_t; s_{t+1} \mid s_t; M) \\
&= \max_{a_t \sim \pi_c(\cdot|s)} \mathcal{H}(\pi_c(a_t|s_t)) - \mathcal{H}(\pi_c(a_t|s_t; s_{t+1})),
\end{aligned}
\tag{5}
$$

where $\pi_c$ is the policy under the causal weighted matrix $M^{a \to r}$. The first entropy term $\mathcal{H}(\pi_c(a_t|s_t))$ promotes action diversity within the constraints of the causal structure. It encourages the agent to explore a wide range of actions that are causally informed, while the second entropy term $-\mathcal{H}(\pi_c(a_t|s_t; s_{t+1}))$ enhances the action predictability in state transitions. It encourages the selection of actions that lead to predictable outcomes, given the current and subsequent states, thereby promoting controlled and goal-oriented behaviors. We train an inverse dynamics model to represent the policy $\pi_c(\cdot|s_t; s_{t+1})$. The detailed derivation proceeds as follows:

$$
\mathcal{H}(\pi_c(\cdot|s_t)) = -\mathbb{E}_{a_t \in \mathcal{A}} \left[ \sum_{i=1}^{d_\mathcal{A}} M^{a^i \to r} \odot \pi_c(a_t^i|s_t) \log \pi(a_t^i|s_t) \right],
\tag{6}
$$

and

$$
\mathcal{H}(\pi_c(\cdot|s_t; s_{t+1})) = -\mathbb{E}_{a_t \in \mathcal{A}} \left[ \sum_{i=1}^{d_\mathcal{A}} M^{a^i \to r} \odot \pi_c(a_t^i|s_t; s_{t+1}) \log \pi(a_t^i|s_t; s_{t+1}) \right],
\tag{7}
$$

where $d_\mathcal{A}$ is the dimension of the action space. Hence, the learning objective of the causal action empowerment can be defined as follows:

$$
\begin{aligned}
\mathcal{E}_{\pi_c}(s) &= \max_{a_t \sim \pi_c(\cdot|s)} \mathcal{H}(\pi_c(a_t|s_t)) - \mathcal{H}(\pi_c(a_t|s_t; s_{t+1})) \\
&= \max_{a_t \sim \pi_c(\cdot|s)} \mathbb{E}_{\pi_c(a_t|s_t)p_{\pi_c}(a_t|s_t, s_{t+1})} \left[ \log \mathcal{P}_{\phi_c}(a_t \mid s_t, s_{t+1}; M) - \log \mathcal{P}_{\pi_c}(a_t|s_t; M) \right],
\end{aligned}
\tag{8}
$$

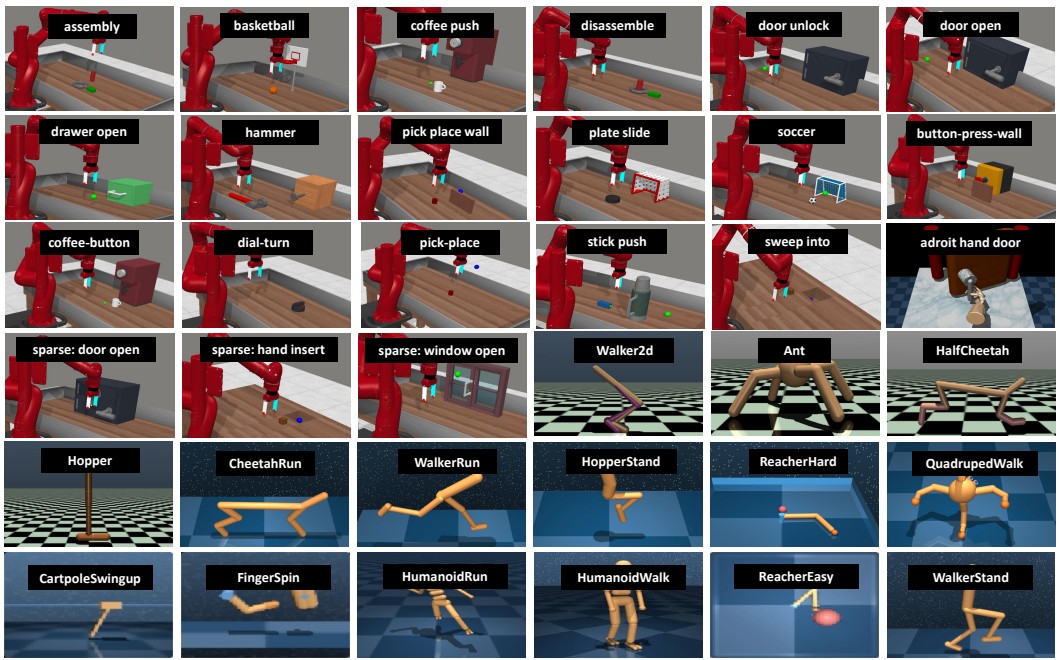

Figure 3: The 36 experimental tasks in 5 continuous control environments

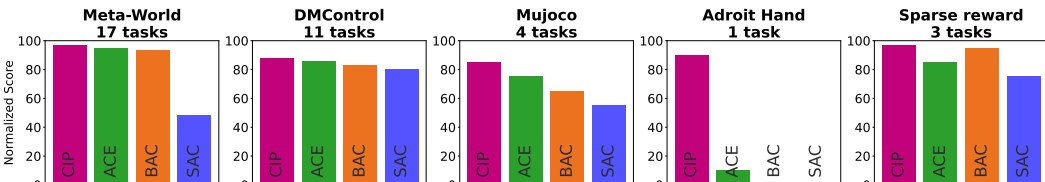

Figure 4: Experimental results with normalized score across all 36 tasks in 5 environments.

where $\mathcal{P}_{\pi_c}(a_t|s_t; M)$ is the action distribution given current state of policy $\pi_c$ with the causal structure, which can be denoted as $\pi_c(a_t|s_t)$. $\mathcal{P}_{\phi_c}(a_t|s_{t+1}, s_t; M)$ represents an inverse dynamics model trained on the collected transitions of state variables. Hence, we update the target policy $\pi_c$ by maximizing the empowerment objective function derived in Eq. 8.

Adhering to the maximum entropy paradigm (Haarnoja et al., 2018), we calculate $\mathcal{E}_{\pi_c}(s)$ for maximization instead of standard entropy, thus prioritizing exploration of pivotal actions that are more likely to have significant causal effects on the reward. This targeted exploration strategy has the potential to accelerate learning by focusing on the most influential actions in current controllable states. Based on the causality-aware empowerment, the Q-value for policy $\pi_c$ could be computed iteratively by applying a modified Bellman operator $\mathcal{T}_c^{\pi}$ with $\mathcal{E}_{\pi_c}(s)$ term as stated below:

$$
\begin{aligned}
\mathcal{T}_c^{\pi} Q(s_t, a_t) &= r(s_t, a_t) + \gamma \mathbb{E}_{s_{t+1} \sim \mathcal{P}} \left[ \mathbb{E}_{a_t \sim \pi_c} \left[ Q(s_{t+1}, a_{t+1}) + \alpha \mathcal{E}_{\phi_c}(s) \right] \right] \\
&= r(s_t, a_t) + \gamma \mathbb{E}_{s_{t+1} \sim \mathcal{P}} \left[ \mathbb{E}_{a_t \sim \pi_c} \left[ Q(s_{t+1}, a_{t+1}) + \alpha (\mathcal{H}(\pi_c(a_t|s_t)) - \mathcal{H}(\pi_c(a_t|s_t; s_{t+1}))) \right] \right].
\end{aligned}
\tag{9}
$$

Hence, we integrate the causality-aware empowerment term into the policy optimization objective function, $\hat{\eta}_{\mathcal{M}}(\pi_c) = \mathbb{E}_{s_0 \sim \mu_0, s_t \sim \mathcal{P}, a_t \sim \pi_c} \left[ \sum_{t=0}^{\infty} \gamma^t (r(s_t, a_t) + \alpha \mathcal{E}_{\pi_c}(s)) \right]$.

In summary, **CIP** harnesses empowerment to integrate the causal understanding into decision-making. By maximizing the empowerment gain of the causally-informed policy, we guide the agent to prioritize actions that align with the environment's underlying causal relationships. This approach enhances the agent's exploration efficiency, focusing on actions with meaningful causal impacts and correlated with desired outcomes. Algorithm 1 illustrates the complete **CIP** pipeline.

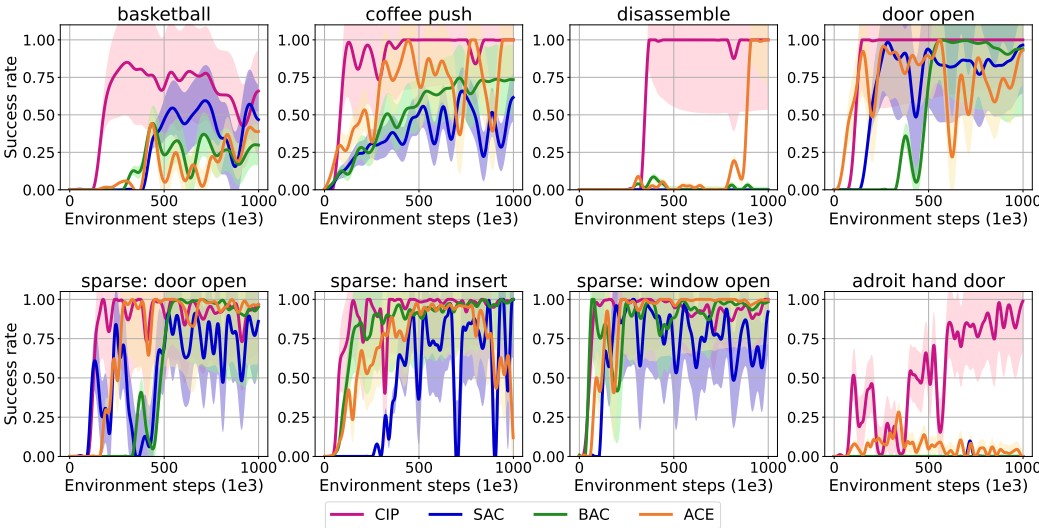

Figure 5: Experimental results of 8 manipulation skill learning tasks in Meta-World and adroit hand environments including sparse reward settings. For all tasks results, please refer to Appendix D.2.

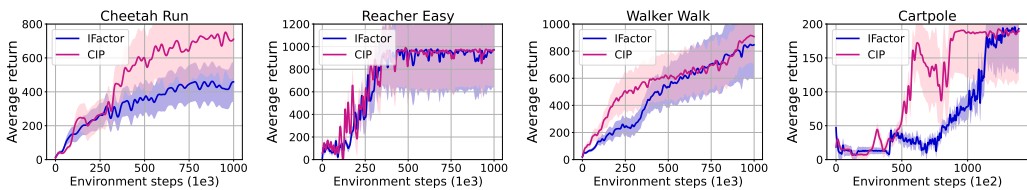

Figure 6: Experimental results of 4 pixel-based tasks in DMControl and Cartpole environments.

## 5 EXPERIMENTS

Our experiments aim to address the following questions: (i) How does the performance of **CIP** compare to other RL approaches in diverse continuous control tasks, including manipulation and locomotion with sparse rewards, high-dimensional action spaces, and pixel-based challenges? (ii) Can **CIP**, through data augmentation and empowerment, improve sample efficiency and learn reliable policies? (iii) What are the effects of the components and hyperparameters in **CIP**?

### 5.1 EXPERIMENTAL SETUP

**Environments.** We evaluate **CIP** on 5 continuous control environments, including MuJoCo (Todorov et al., 2012), DMControl (Tassa et al., 2018), Meta-World (Yu et al., 2020), Adroit Hand (Rajeswaran et al., 2018), and sparse reward setting environments in Meta-World. This comprehensive evaluation encompasses 36 tasks, spanning both locomotion and manipulation skill learning, as illustrated in Figure 3. We also conduct experiments in 4 pixel-based tasks of the DMControl and Cartpole environment as shown in Figure 17. Our experimental tasks incorporate a wide range of challenges, including high-dimensional state and action spaces, sparse reward settings, pixel-based scenarios, and locomotion. For extensive experimental settings, please refer to Appendix D.1.

**Baselines.** We compare **CIP** with three popular RL baselines across all 36 tasks and against IFactor (Liu et al., 2024) in 3 pixel-based tasks: (1) SAC (Haarnoja et al., 2018), an off-policy actor-critic algorithm featuring maximum entropy regularization. (2) ACE (Ji et al., 2024a), a method employing causality-aware entropy regularization. (3) BAC (Ji et al., 2024b). a method that balances sufficient exploitation of past successes with exploration optimism. (4) IFactor (Liu et al., 2024). a causal framework modeling four distinct categories of latent state variables for pixel-based tasks. To ensure robustness and statistical significance, we conduct each experiment using 4 random seeds.

### 5.2 MAIN RESULTS

Figure 4 presents the normalized scores of **CIP** compare to other methods across 36 tasks in 5 environments. In 17 Meta-World robot-arm tasks, **CIP** achieves a near-perfect score of 100,

Table 1: The experimental results of average return in 8 locomotion tasks. We bold the best scores, and underline second-best results, $\pm$ is the standard deviation, w/o represents without. ● indicates **CIP** is statistically superior to compared method (pairwise $t$-test at $95\%$ confidence interval).

| Method | Ant | HalfCheetah | Hopper | Walker2d | Cheetah Run | Hopper Stand | Quadruped Walk | Reacher Hard |
|---|---|---|---|---|---|---|---|---|
| **CIP** | 6418±81 | **12594±210** | **2846±882** | **5624±91** | **893±12** | **936±17** | 948±54 | **991±11** |
| **w/o Aug** | 6231±81 | 12225±102 | 2308±785 | 5294±41 | 885±13 | 931±22 | 945±35 | 989±13 |
| **w/o Emp** | 6295±210 | 10986±572 | 2270±904 | 5547±91 | 876±21 | 785±114 | 924±23 | 971±13 |
| SAC | 6062±105● | 10888±240● | 2266±981 | 5251±106 | 767±16● | **936±8** | 930±19 ● | 980±8● |
| BAC | **6511±30** | 10276±34● | 2263±1063● | 3316±702 ● | 665±6● | 932±4● | **962±24** | 974±16 |
| ACE | 5922±106 ● | 9390±25● | 2312±673● | 4922±96● | 863±23● | 912±16 | 933±57 | 973±17 |

showcasing its exceptional performance in manipulation tasks. For locomotion tasks in DMControl and MuJoCo, **CIP** consistently attains scores exceeding 80, indicating robust performance across diverse locomotion challenges. Notably, **CIP** exhibits significant performance improvements in challenging scenarios, such as adroit hand manipulation and 3 tasks with the sparse reward setting. These results underscore the effectiveness in tackling complex, high-dimensional control problems. In next sections, we present a comprehensive analysis of **CIP**'s performance across diverse tasks.

**Robot-arm manipulation.** Figure 5 presents the success rates across 7 Meta-World robot-arm manipulation tasks including sparse reward settings. **CIP** consistently outperforms all other methods across these tasks, demonstrating both faster policy learning and enhanced stability. In challenging tasks, such as disassemble, **CIP** achieves an impressive 100% success rate. The effectiveness of **CIP** can be attributed to focus on causally relevant information within the state and action spaces. In sparse reward settings, the efficient extraction of causal state information and the prioritization of controllable actions enable effective task completion. By systematically eliminating noise from non-causal factors, **CIP** allows the agent to construct a more controllable and efficient policy.

**High-dimensional Adroit hand manipulation.** To rigorously evaluate our method's efficacy in high-dimensional tasks, we conduct comparative experiments in the Adroit Hand environment of door open task. This challenging setup involves controlling a robotic hand with up to 28 actuated degrees of freedom ($\mathcal{A} \in \mathbb{R}^{28}$). Figure 5 illustrates the success rates achieved across all methods. Notably, while the three other comparative methods fail to demonstrate significant progress on this challenging task, **CIP** achieves a near $100\%$ success rate after $700k$ environment steps.

**Locomotion.** We further evaluate **CIP** in another important category: locomotion. The part experimental results of average return in MuJoCo and DMControl environments are presented in Table 1. Learning curves are illustrated in Figure 7. We observe that **CIP** achieves the best performance in six tasks and sub-optimal in other tasks, and shows statistically significant improvements in 5 out of 8 tasks. Moreover, compared to the traditional method SAC, **CIP** demonstrates significant performance improvements in more challenging tasks such as CheetahRun and Hopper. Compared to the causality-based method ACE, **CIP** demonstrates improvements in all tasks. Overall, in locomotion tasks, **CIP** achieves superior performance and attains high sample efficiency. Detailed performance and statistical analyses are provided in Appendix D.2 and D.3.5.

**Pixel-based task learning** To further validate the performance in pixel-based tasks, we use 3 complex pixel-based DMControl tasks for evaluation, where video backgrounds serve as distractors. We apply the proposed counterfactual data augmentation and causal action empowerment to IFactor for comparison. As shown in Figure 6, **CIP** surpasses IFactor in terms of average return. These results underscore **CIP**'s efficacy in pixel-based tasks and its capacity to better overcome spurious correlations arising from video backgrounds, focusing on locomotion. Moreover, the result of Cartpole task in Figure 6 demonstrate the effectiveness in discrete action space environment. For visualization trajectories in pixel-based results, please refer to Appendix D.2.4.

### 5.3 ANALYSIS

**Ablation study.** We conduct ablation experiments involving **CIP**, **CIP** without (w/o) counterfactual data augmentation (Aug), and **CIP** w/o Empowerment (Emp). The results in 8 locomotion tasks are shown in Table 1. And the learning curves of all tasks are depicted in Appendix D.3. The

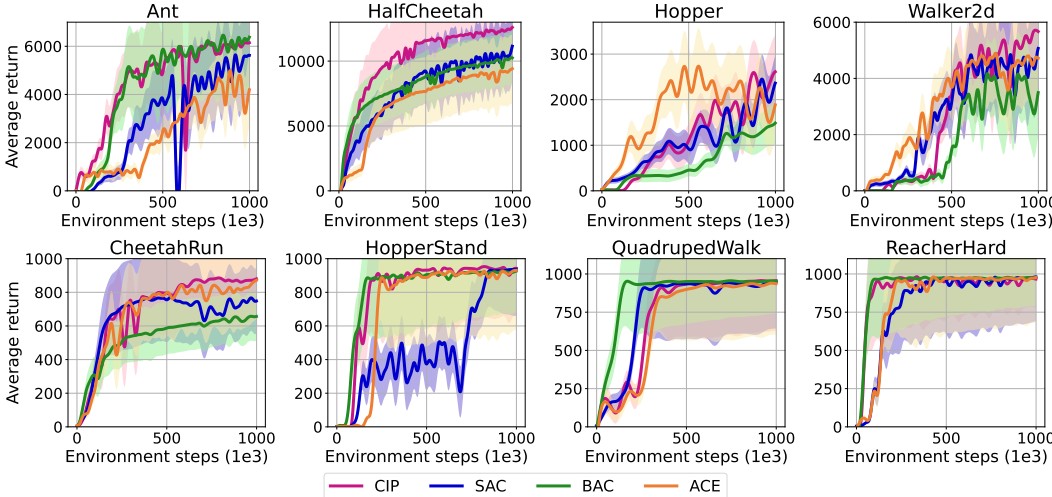

Figure 7: Experimental results with average return across 8 tasks in locomotion tasks.

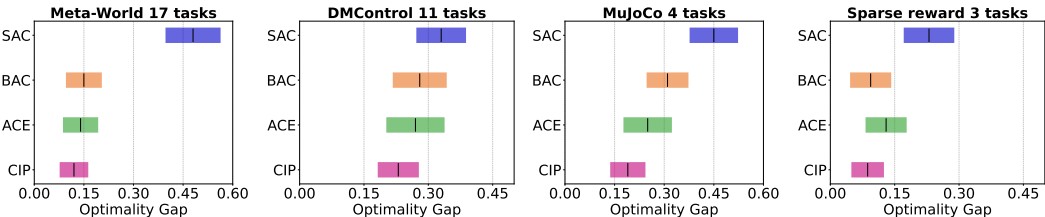

Figure 8: Experimental results of reliability evaluation by the metric Optimality Gap (lower values are better) on 4 diverse environments across 35 tasks.

experimental results reveal that the variant without the empowerment learning objective performs poorly, underscoring the critical role of empowerment maximization in enhancing control capabilities. Additionally, **CIP** without counterfactual data augmentation is less sample efficient than **CIP**, highlighting the importance of augmentation.

**Reliability evaluation.** We evaluate **CIP**'s reliability across 35 tasks in 4 environments, excluding the Adroit Hand door task due to **CIP**'s exceptional performance there. Figure 8 illustrates the experimental results using the Optimality Gap metric (Agarwal et al., 2021). **CIP** consistently achieves the lowest values across all tasks in four environments, with lower values indicating superior performance. This consistent excellence across diverse scenarios underscores the robustness and reliability of our proposed method.

## 6 CONCLUSION

This study introduces an efficient RL framework, designed to enhance sample efficiency. This approach begins by counterfactual data augmentation using the causality between states and rewards, effectively mitigating interference from irrelevant states without additional environmental interactions. We then develop a reward-guided structural model that leverages causal awareness to prioritize causal actions through empowerment. We conduct extensive experiments across 39 tasks spanning 5 diverse continuous control environments which demonstrate the exceptional performance of our proposed method, showcasing its robustness and adaptability across challenging scenarios.

**Limitation and Future Work** The current limitations of our work are twofold. First, **CIP** has not yet been extended to complex scenarios, such as real-world 3D robotics tasks. Potential approaches to address this limitation include leveraging object-centric models (Wu et al., 2023), 3D perception models (Wang et al., 2024a), and robotic foundation models (Team et al., 2024; Firoozi et al., 2023) to construct essential variables for causal world modeling. Second, **CIP** does not adequately consider non-stationarity and heterogeneity, which are critical challenges in causal discovery. Future work could integrate method designed to handle such complexities, such as CD-NOD (Huang et al., 2020).

## REPRODUCIBILITY STATEMENT

We provide the core code of **CIP** in the supplementary material. The implementation details are shown in Appendix D.1.

## ACKNOWLEDGMENT

This work was supported in part by the National Natural Science Foundation of China under Grant 62276128, Grant 62192783; in part by the Jiangsu Science and Technology Major Project BG2024031; in part by the Natural Science Foundation of Jiangsu Province under Grant BK20243051; the Fundamental Research Funds for the Central Universities(14380128); in part by the Collaborative Innovation Center of Novel Software Technology and Industrialization.

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

CONTENTS

## A    BROADER IMPACT

To avoid blind exploration and improve sample efficiency, we propose `CIP` for efficient reinforcement learning. `CIP` leverages the causal relationships among states, actions, and rewards to prioritize causal information for efficient policy learning. `CIP` first learns a causal matrix between states and rewards to execute counterfactual data augmentation, prioritizing important state features without additional environmental interactions. Subsequently, it learns a causal reweight matrix between actions and rewards to prioritize causally-informed behaviors. We then introduce a causal action empowerment term into the learning objective to enhance the controllability. By prioritizing the causal information, `CIP` enables agents to focus on behaviors that have causally significant effects on their tasks. `CIP` offers substantial broader impact by prioritizing causal information through individual assessment of how different factors contribute to rewards. Our novel empowerment learning objective achieves efficient policy optimization by leveraging entropy via the policy and learned inverse dynamics model. This approach shows promise for extension into research frameworks centered on maximum entropy algorithms.

Despite its strengths, `CIP` has limitations beyond its reliance on the method DirectLiNGAM. There's potential to explore alternative causal discovery techniques for more robust relationship mapping. Moreover, analyzing inter-entity causal connections could lead to better disentanglement of diverse behaviors. Our future work will investigate a range of causal discovery methods to refine our approach. We aim to extend `CIP` to model-based RL frameworks, focusing on building causal world models to enhance generalization.

## B    ASSUMPTIONS AND PROPOSITIONS

**Assumption 1** *(d-separation (Pearl, 2009)) d-separation is a graphical criterion used to determine, from a given causal graph, if a set of variables X is conditionally independent of another set Y, given a third set of variables Z. In a directed acyclic graph (DAG) $\mathcal{G}$, a path between nodes $n_1$ and $n_m$ is said to be blocked by a set S if there exists a node $n_k$, for $k = 2, \cdots, m-1$, that satisfies one of the following two conditions:*

*(i) $n_k \in S$, and the path between $n_{k-1}$ and $n_{k+1}$ forms $(n_{k-1} \to n_k \to n_{k+1})$, $(n_{k-1} \leftarrow n_k \leftarrow n_{k+1})$, or $(n_{k-1} \leftarrow n_k \to n_{k+1})$.*

*(ii) Neither $n_k$ nor any of its descendants is in S, and the path between $n_{k-1}$ and $n_{k+1}$ forms $(n_{k-1} \to n_k \leftarrow n_{k+1})$.*

*In a DAG, we say that two nodes $n_a$ and $n_b$ are d-separated by a third node $n_c$ if every path between nodes $n_a$ and $n_b$ is blocked by $n_c$, denoted as $n_a \perp\!\!\!\perp n_b | n_c$.*

**Assumption 2** *(Global Markov Condition (Spirtes et al., 2001; Pearl, 2009)) The state is fully observable and the dynamics is Markovian. The distribution p over a set of variables $\mathcal{V} = (s_t^1, \cdots, s_t^d, a_t^1, \cdots, a_t^d, r_t)^T$ satisfies the global Markov condition on the graph if for any partition $(\mathcal{S}, \mathcal{A}, \mathcal{R})$ in $\mathcal{V}$ such that if $\mathcal{A}$ d-separates $\mathcal{S}$ from $\mathcal{R}$, then $p(\mathcal{S}, \mathcal{R}|\mathcal{A}) = p(\mathcal{S}|\mathcal{A}) \cdot p(\mathcal{R}|\mathcal{A})$*

**Assumption 3** *(Faithfulness Assumption (Spirtes et al., 2001; Pearl, 2009)) For a set of variables $\mathcal{V} = (s_t^1, \cdots, s_t^d, a_t^1, \cdots, a_t^d, r_t)^T$, there are no independencies between variables that are not implied by the Markovian Condition.*

**Assumption 4** *Under the assumptions that the causal graph is Markov and faithful to the observations, the edge $s_t^i \to s_{t+1}^i$ exists for all state variables $s^i$.*

**Assumption 5** *No simultaneous or backward edges in time.*

**Proposition 1** *Under the assumptions that the causal graph is Markov and faithful to the observations, there exists an edge from $a_t^i \to r_t$ if and only if $a_t^i \not\perp\!\!\!\perp r_t | a_t \setminus a_t^i, s_t$.*

*Proof:* We proceed by proving both directions of the if and only if statement.

($\Rightarrow$) Suppose there exists an edge from $a_t^i$ to $r_t$. We prove that $a_t^i \not\perp\!\!\!\perp r_t | a_t \setminus a_t^i, s_t$ by contradiction. Assume $a_t^i \perp\!\!\!\perp r_t | a_t \setminus a_t^i, s_t$. By the faithfulness assumption, this independence must be reflected in

the graph structure. However, this implies the absence of a directed path from $a_t^i$ to $r_t$, contradicting the existence of the edge. Thus, $a_t^i \not\perp\!\!\!\perp r_t | a_t \setminus a_t^i, s_t$.

($\Leftarrow$) Now, suppose $a_t^i \not\perp\!\!\!\perp r_t | a_t \setminus a_t^i, s_t$. We prove the existence of an edge from $a_t^i$ to $r_t$ by contradiction. Assume no such edge exists. By the Markov assumption, the absence of this edge implies $a_t^i \perp\!\!\!\perp r_t | a_t \setminus a_t^i, s_t$, contradicting our initial supposition. Therefore, an edge from $a_t^i$ to $r_t$ must exist. Thus, we have shown that an edge from $a_t^i$ to $r_t$ exists if and only if $a_t^i \not\perp\!\!\!\perp r_t | a_t \setminus a_t^i, s_t$, completing the proof.

**Proposition 2** *Under the assumptions that the causal graph is Markov and faithful to the observations, there exists an edge from $s_t^i \to r_t$ if and only if $s_t^i \not\perp\!\!\!\perp r_t | \{a_t, s_t \setminus r_t\}$.*

The proof of Proposition 2 follows a similar line of reasoning as that of Proposition 1.

**Theorem 1** *Based on above 5 assumptions and 2 propositions, suppose $s_t, a_t, s_t$ follow the factored MDP reward function Eq. 2, the causal matrices $M^{s \to r}$ and $M^{a \to r}$ are identifiable.*

## C  EXTENDED RELATED WORK

We categorize existing causal RL approaches based on problem domains and task types, providing a systematic analysis of how different methods explore causal relationships between states, actions, and rewards, as illustrated in Table 2.

In the single-task learning domain, methods such as ACE (Ji et al., 2024a) and IFactor (Liu et al., 2024) have shown success in learning policies for manipulation and locomotion tasks. However, both approaches are limited by focusing on a single reward-guided causal relationship. Regarding generalization, AdaRL (Huang et al., 2022a) effectively leverages both state-reward and action-reward causal relationships. However, AdaRL focuses primarily on applying causal inference to address generalization challenges in locomotion tasks. Its application is limited to locomotion tasks, leaving more complex manipulation tasks unaddressed. Since our work focuses on the single-task problem domain, we do not provide a direct comparison with AdaRL. Conversely, CBM (Wang et al., 2024d) considers the causal relationship between states and rewards but overlooks the causal link between actions and rewards. In the problem domain of counterfactual data augmentation, current causal RL methods (Urpí et al., 2024; Pitis et al., 2020; 2022) have not yet explored the inference and utilization of both causal relationships.

In summary, current research on reward-guided causal discovery remains incomplete and lacks validation across a broader spectrum of tasks. This gap underscores the need for more comprehensive investigation and application in the field of causal reinforcement learning.

### C.1  EXTENDED DISCUSSION ON OBJECT-CENTRIC RL AND 3D WORLD MODELS

The main similarity lies between our framework and object-centric RL is both are learning and using factored MDPs (Kearns & Koller, 1999), but they differ in granularity: our framework operates at the component level (e.g., raw state variables), whereas object-centric RL factors states based on objects.

Although our work is orthogonal to object-centric RL, we believe certain elements of object-centric RL could complement our framework in specific applications, particularly in real-world robotic manipulation tasks. Potential future work include:

- **Using object-centric representation as input**: Object-centric models can help identify object-factored variables, such as object attributes, geometry, and physical states, which are useful for planning (Jiang et al., 2019; Lin et al., 2020; Kossen et al., 2019; Mambelli et al., 2022; Feng & Magliacane, 2023; Choi et al., 2024; Zadaianchuk et al., 2022; Park et al., 2021; Zadaianchuk et al., 2021; Yuan et al., 2022; Li et al., 2020; Mitash et al., 2024; Haramati et al., 2023; Li et al., 2024). In this case, states are factored as objects, and we can learn causal graphs over these variables. This is useful in robotic environments involving numerous objects. We will leave this as a future work for adapting our current framework to the applications of the object-centric robotic task.

Table 2: Categorization of different causal RL methods with two different causal relationship of state-to-reward (state-reward) and action-to-reward (action-reward).

| Problem domain | Task type | Method | Causal relationship | |
|---|---|---|---|---|
| | | | state-reward | action-reward |
| **Single-task** | manipulation; locomotion | ACE (Ji et al., 2024a) | ✗ | ✓ |
| | manipulation; locomotion | IFactor (Liu et al., 2024) | ✓ | ✗ |
| | manipulation | CAI (Seitzer et al., 2021) | ✗ | ✗ |
| **Generalization** | manipulation | CDL (Wang et al., 2022) | ✗ | ✗ |
| | locomotion | AdaRL (Huang et al., 2022a) | ✓ | ✓ |
| | manipulation; locomotion | CBM (Wang et al., 2024d) | ✓ | ✗ |
| **Augmentation** | manipulation | CAIAC (Urpí et al., 2024) | ✗ | ✗ |
| | manipulation | CoDA (Pitis et al., 2020) | ✗ | ✗ |
| | manipulation | MoCoDA (Pitis et al., 2022) | ✗ | ✗ |

- **Learning more compact factored object representations with our framework**: Our structure learning approach could benefit object-centric RL by disentangling the internal representations of individual objects to the reward-relevant and reward-irrelevant groups by learning the causal structures. This can enhance the compactness and interpretability of object-centric representations.

- **Using object-aware 3D world models for applications**: In 3D environments, object-aware 3D world models (Li & Pathak, 2024) can provide essential representations of objects. Our framework could then build causal structures on top of these factored 3D-object representations.

While these directions are promising and could advance the applicability of our framework in certain domains, they are outside the primary focus of this work. We plan to explore these ideas as part of future work.

## D  DETAILS ON EXPERIMENTAL DESIGN AND RESULTS

### D.1  EXPERIMENTAL SETUP

We present the detailed hyperparameter settings of the proposed method `CIP` across all 5 environments in Table 3. Additionally, the Q-value and V-value networks are used MLP with 512 hidden size. And the policy network is the Gaussian MLP with 512 hidden size. Moreover, we set the target update interval of 2. For fair comparison, the hyperparameters of the baseline methods (SAC (Haarnoja et al., 2018), BAC (Ji et al., 2024b), ACE (Ji et al., 2024a)) follow the same settings in the experiments.

For pixel-based DMControl environments, we employ IFactor (Liu et al., 2024) to encode latent states and integrate the `CIP` framework for policy learning. We utilize the $s_t^{\bar{r}}$ state features in IFactor as uncontrollable states unrelated to rewards to execute counterfactual data augmentation. Furthermore, for simplicity, we maximize the mutual information between future states and actions to facilitate empowerment. All parameter settings in these three tasks adhere to those specified in IFactor. Additionally, We use the same background video for the comparison.

### D.2  FULL RESULTS

#### D.2.1  EFFECTIVENESS IN ROBOT ARM MANIPULATION

Figure 9 presents the learning curves for all 17 manipulation skill tasks within the Meta-World environment. The `CIP` framework demonstrates superior learning outcomes and efficiency compared to the three baseline methods, despite exhibiting minor instabilities in the basketball and dial-turn tasks. Notably, `CIP` achieves a 100% success rate in more complex tasks, such as pick-place-wall and assembly. The visualization results presented in Figures 11 and 12 further demonstrate `CIP`'s

Table 3: Hyperparameter settings of `CIP` in 5 environments

| Hyperparameter | Environment | | | | |
|---|---|---|---|---|---|
| | Meta-World | Sparse | MuJoCo | DMControl | Adroit Hand |
| batch size | 512 | 512 | 256 | 512 | 256 |
| hidden size | 1024 | 1024 | 256 | 1024 | 256 |
| Q-value network hidden size | | | 512 | | |
| V-value network hidden size | | | 512 | | |
| policy network hidden size | | | 512 | | |
| learning step | | | 1000000 | | |
| replay size | | | 1000000 | | |
| causal sample size | | | 10000 | | |
| gamma | | | 0.99 | | |
| learning rate | | | 0.0003 | | |
| update interval | | | 2 | | |

ability to effectively and efficiently complete tasks, even in high-dimensional action spaces such as the Adroit Hand environment.

In the hammer task, `CIP` allows the robot arm to execute reach and pick actions with precision, enabling it to accurately identify the nail's position and successfully perform the hammering action. In the Adroit Hand door task, `CIP` effectively controls the complex joints to grasp the doorknob and applies the appropriate force to twist it, thereby opening the door.

These findings affirm the effectiveness of `CIP` in robot arm manipulation skill learning, highlighting its capacity to enhance sample efficiency while mitigating the risks associated with blind exploration.

**Visualization.**  We employ trajectory visualization to comparatively validate the efficacy of our method. As depicted in Figure 10, the light-shaded regions delineate the policy exploration space, while the point clustering area indicates the area of frequent interaction. Our analysis reveals that `CIP`, leveraging counterfactual data augmentation, achieves substantially broader exploration compared to ACE and SAC. Concurrently, the causal information prioritization framework facilitates more focused execution in critical state regions. These visual findings provide robust empirical support for the effectiveness of our proposed augmentation framework.

### D.2.2  EFFECTIVENESS IN SPARE REWARD SETTINGS

Figure 13 presents the learning curves for all three sparse reward setting tasks within the Meta-World environment, while Figure 14 showcases their corresponding visualization trajectories. These findings reveal that `CIP` not only achieves superior learning efficiency but also adeptly executes critical actions necessary for task completion, such as opening the door and window and maneuvering the node to the target place.

These results substantiate the effectiveness of `CIP` in sparse reward scenarios. The counterfactual data augmentation process prioritizes salient state information, effectively filtering out irrelevant factors that could hinder learning. Meanwhile, causal action empowerment enhances policy controllability by focusing on actions that are causally linked to desired outcomes. This dual approach not only accelerates the learning process but also fosters a more robust policy capable of navigating the complexities inherent in sparse reward settings. Overall, these findings underscore `CIP`'s potential to significantly improve performance in challenging environments characterized by limited feedback.

### D.2.3  EFFECTIVENESS IN LOCOMOTION

We further evaluate `CIP` in 15 locomotion tasks in DMControl and MuJoCo environments. Figure 15 presents the learning curves, while Figure 16 showcases the corresponding visualization trajectories in 4 specific tasks. A comprehensive analysis indicates that `CIP` achieves faster learning efficiency and greater stability compared to ACE and SAC, while demonstrating comparable policy learning

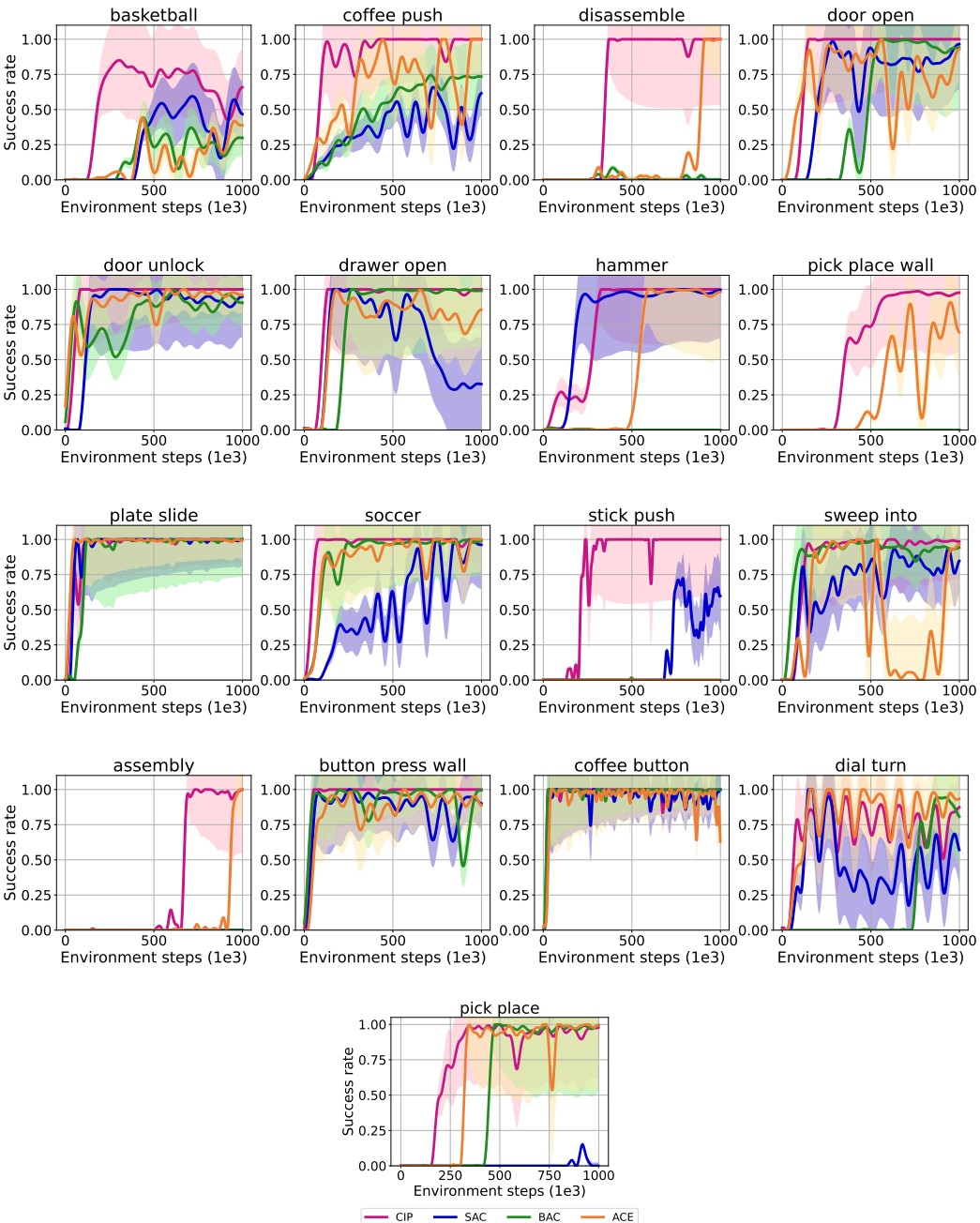

Figure 9: Experimental results across 17 manipulation skill learning tasks in Meta-World.

performance to BAC, which is known for its proficiency in control tasks. The visualization results reveal that **CIP** effectively executes running and walking actions in complex humanoid scenarios.

These findings collectively underscore the efficacy of **CIP** in locomotion tasks, highlighting its potential to advance the state-of-the-art in reinforcement learning for intricate motor control problems. The method's success across varied environments suggests a robust framework that could generalize effectively to other challenging domains within robotics and control systems, paving the way for future research and applications in these areas.

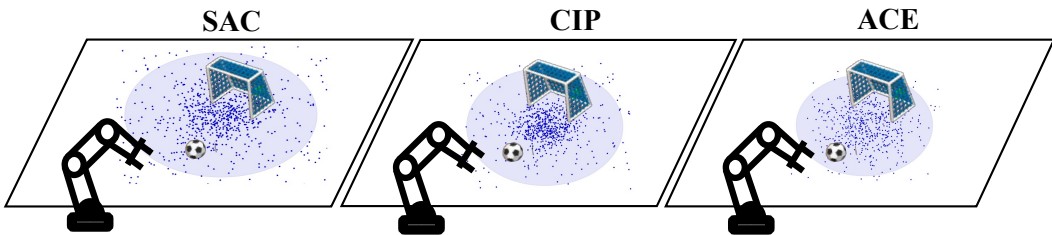

Figure 10: Visualization of the trajectories in soccer task.

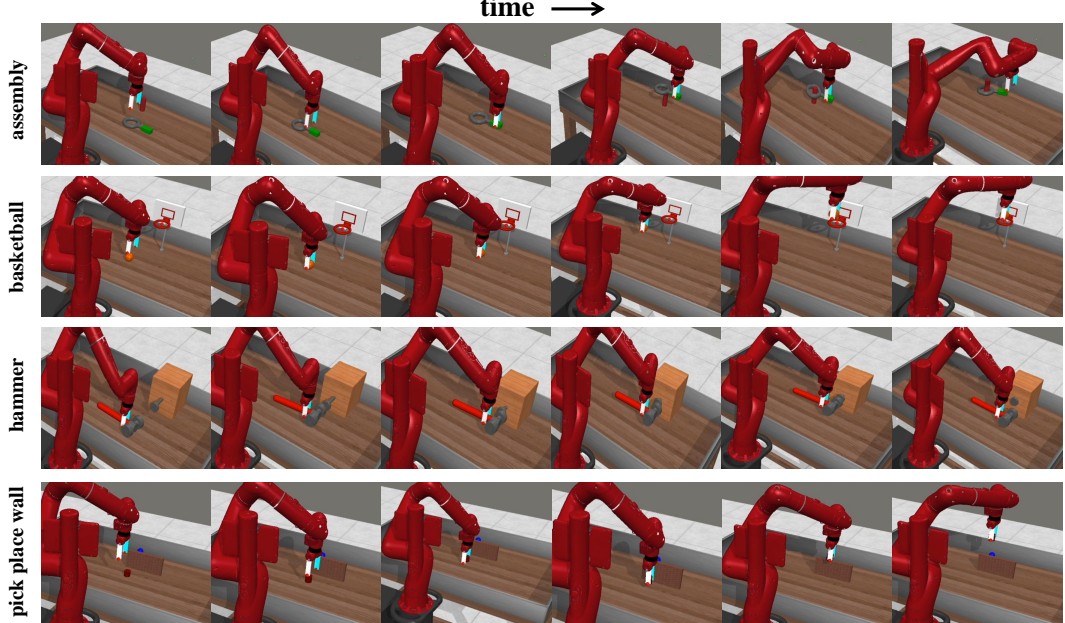

Figure 11: Visualization trajectories of 4 manipulation skill learning tasks in Meta-World environment.

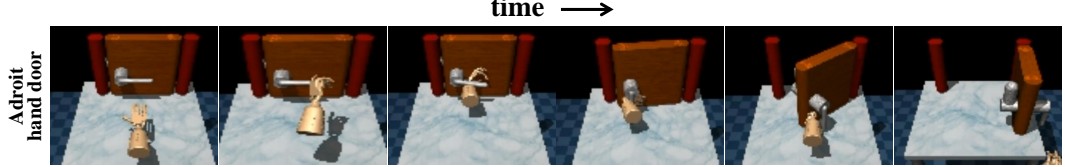

Figure 12: Visualization trajectory of Adroit Hand door open task.

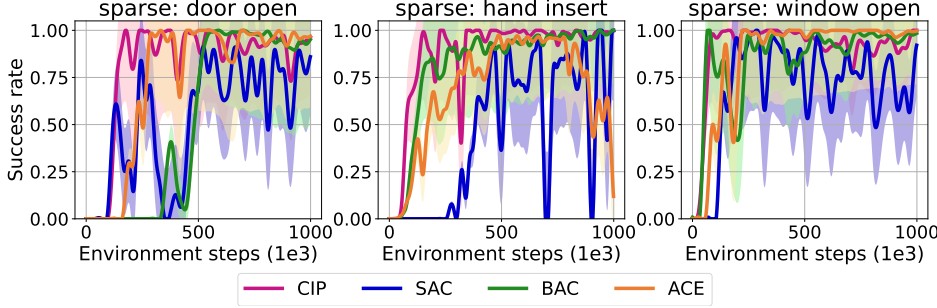

Figure 13: Experimental results across 3 manipulation skill learning tasks in sparse reward settings of Meta-World environment.

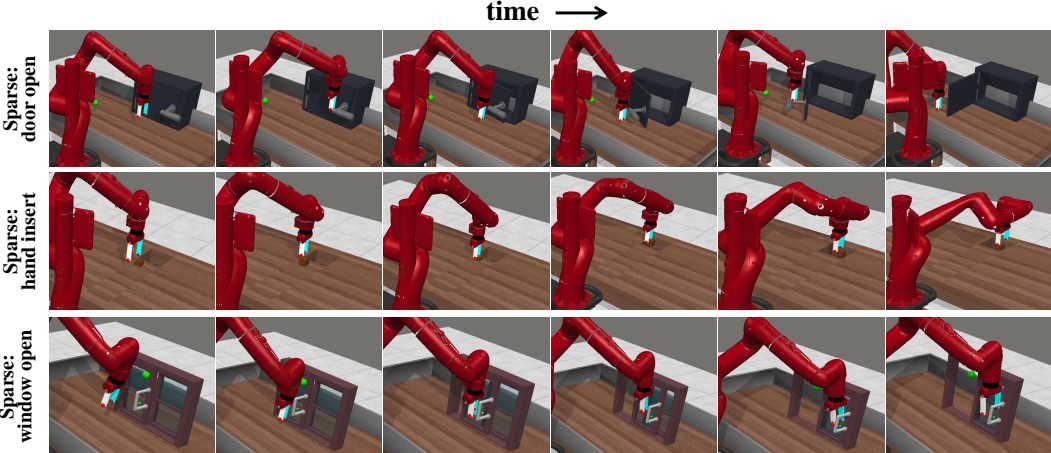

Figure 14: Visualization trajectories of 3 manipulation skill learning tasks in sparse reward settings of Meta-World environment.

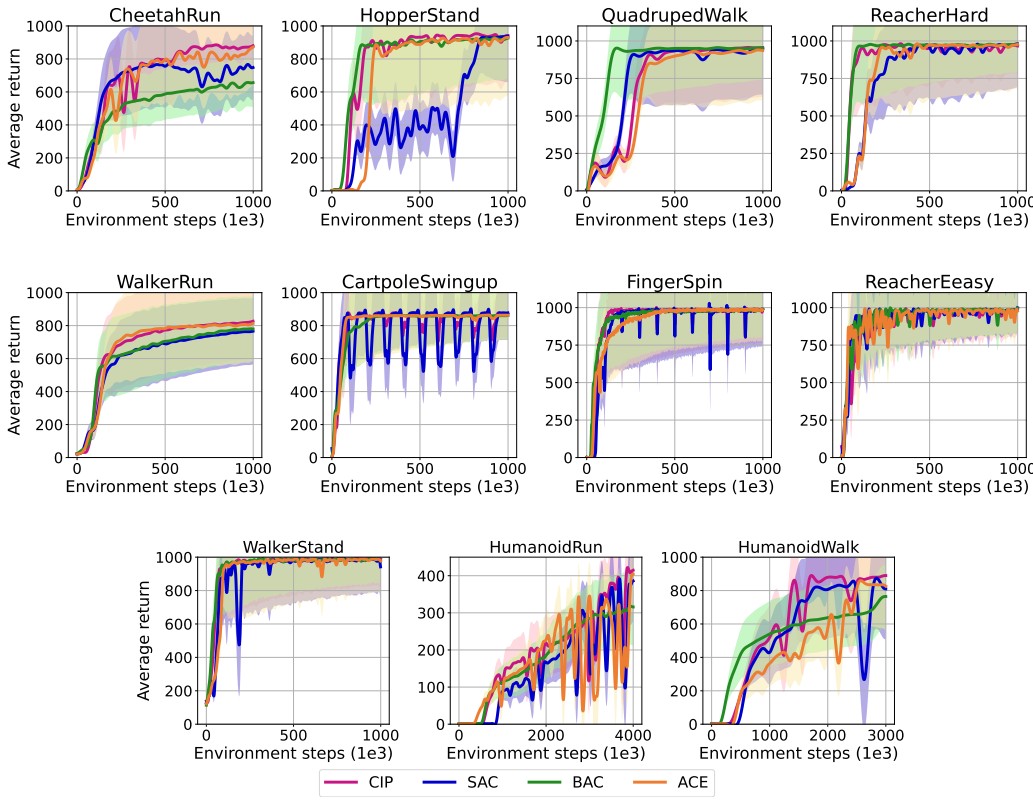

Figure 15: Experimental results across 11 locomotion tasks in DMControl environment.

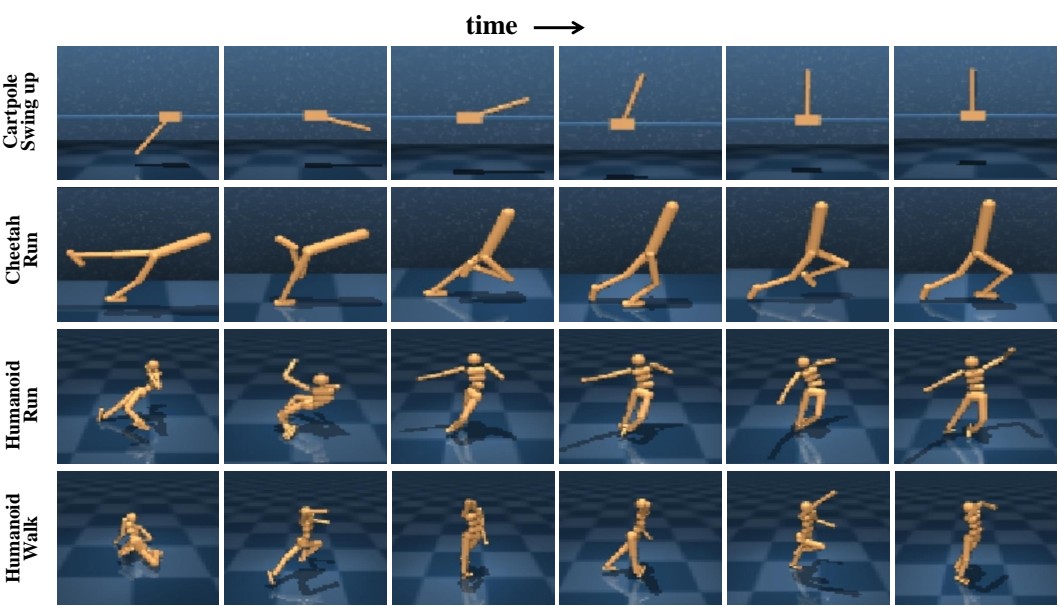

Figure 16: Visualization trajectories of 4 locomotion tasks in DMControl environment.

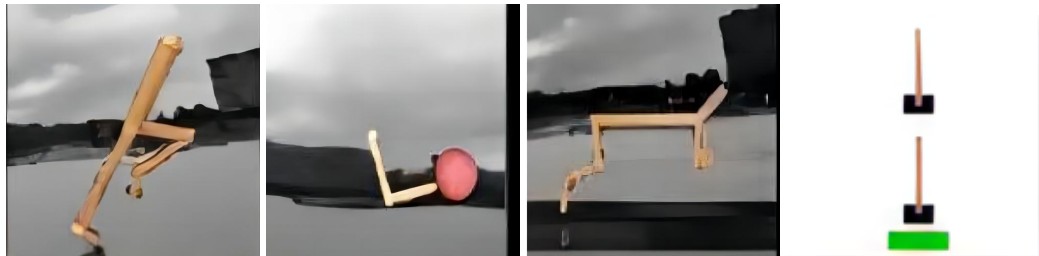

Figure 17: The DMControl environment of 3 pxiel-based tasks (Walker Walk, Cheetah Run, Reacher Easy) and 1 task in Cartpole environment (Liu et al., 2024).

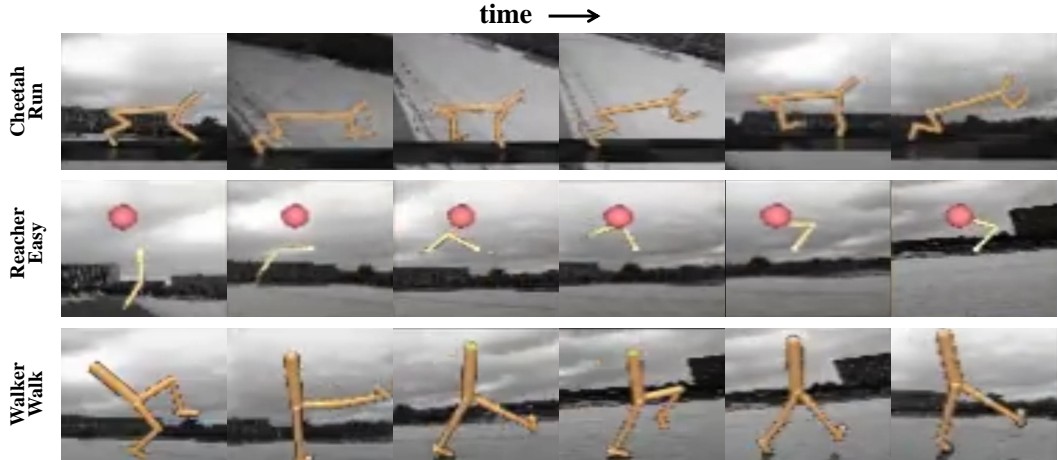

Figure 18: Visualization trajectories in 3 pixel-based locomotion tasks of DMControl environment with video backgrounds as distractors.

### D.2.4 EFFECTIVENESS IN PIXEL-BASED TASKS

To further validate the effectiveness of our proposed framework in pixel-based environments, we evaluated **CIP** on three DMControl pixel-based tasks. We leverage IFactor for latent state processing and differentiation of uncontrollable state features to execute counterfactual data augmentation, alongside maximizing the mutual information between future states and actions for empowerment.

Figure 6 presents the learning curves, while Figure 18 shows the visualization trajectories. The proposed framework exhibits enhanced policy learning performance and effectively mitigates interference from background video, facilitating efficient locomotion. These findings reinforce the effectiveness and extensibility of our causal information prioritization framework, highlighting its potential to improve learning in complex, pixel-based environments.

### D.3 PROPERTY ANALYSIS

### D.3.1 ANALYSIS FOR REPLACING COUNTERFACTUAL DATA AUGMENTATION

In **CIP**, we exploit the causal relationship between states and rewards to perform counterfactual data augmentation on irrelevant state features, thus prioritizing critical state information. We compare this approach with an alternative method: masking irrelevant state features to achieve state abstraction for subsequent causal action empowerment and policy learning. To evaluate the efficacy of both approaches, we conduct experiments with **CIP** with counterfactual data augmentation (**CIP** w/i Cda) and **CIP** with causally-informed states (**CIP** w/i Cs) across three distinct environments.

Figure 19 illustrates comparative results for four manipulation skill learning tasks in the Meta-World environment. Both **CIP** variants achieve 100% task success rates with high sample efficiency, validating their effectiveness. Notably, **CIP** w/i Cda exhibits superior learning efficiency compared to

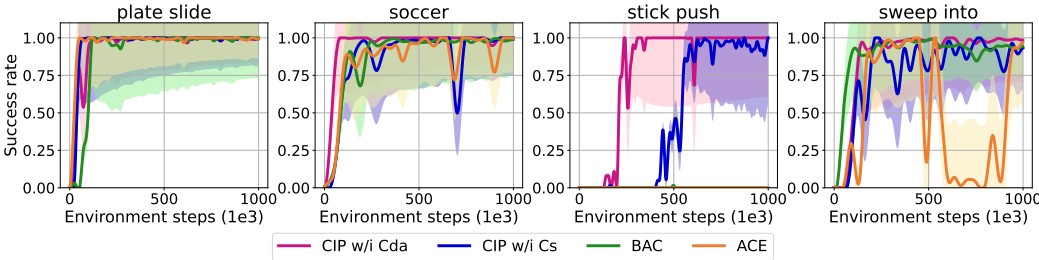

Figure 19: Experimental results in 4 manipulation skill learning tasks of Meta-World environment. w/i stands for with.

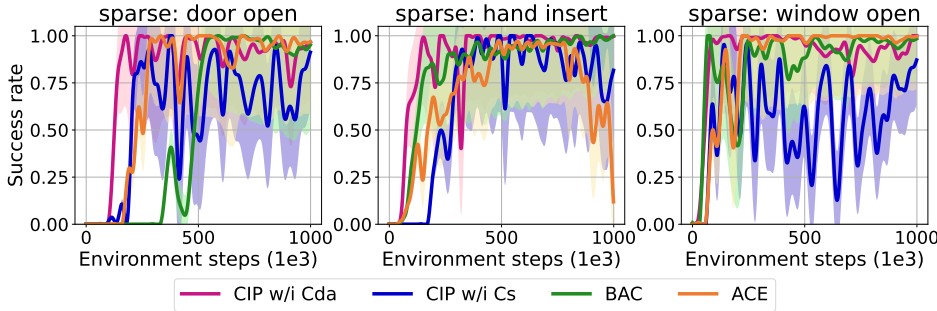

Figure 20: Experimental results in 3 manipulation skill learning tasks of Meta-World environment with sparse reward settings.

**CIP** w/i Cs, underscoring the value of our counterfactual data augmentation approach in enhancing training data without additional environmental interactions. In three sparse reward setting tasks (Figure 20), **CIP** w/i Cda demonstrates superior policy performance. Further experiments across four locomotion environmetal tasks corroborate these findings, consistently favoring the counterfactual data augmentation approach. These comprehensive experimental results strongly support the effectiveness and significance of incorporating counterfactual data augmentation in **CIP**, highlighting its potential to enhance reinforcement learning across diverse task domains.

### D.3.2 EXTENSIVE ABLATION STUDY

**Robot arm manipulation** The ablation study results in the Meta-World and Adroit Hand environments are presented in Figure 22. The findings indicate that **CIP** without counterfactual data augmentation exhibits reduced learning efficiency and is unable to successfully complete tasks such as pick-and-place. This underscores the importance of incorporating counterfactual data augmentation, which prioritizes causal state information, to enhance learning efficiency by mitigating the influence of irrelevant state information and preventing policy divergence.

Furthermore, **CIP** without causal action empowerment demonstrates a significant decline in policy performance across robot arm manipulation tasks. In complex scenarios, such as Adroit Hand door opening and assembly, it fails to learn effective strategies for task completion. This outcome further corroborates the efficacy of the proposed causal action empowerment mechanism, as prioritizing causally informed actions facilitates more efficient exploration of the environment, ultimately enabling successful policy learning.

**Sparse reward settings** Figure 22 presents the results of the ablation study conducted across three sparse reward setting tasks. These findings underscore the substantial influence of causal action empowerment on the efficacy of policy learning, demonstrating its critical role in enhancing performance in challenging environments. Additionally, the incorporation of counterfactual data augmentation proves effective in mitigating the need for additional environmental interactions, thereby significantly improving sample efficiency. This approach not only facilitates more rapid learning but

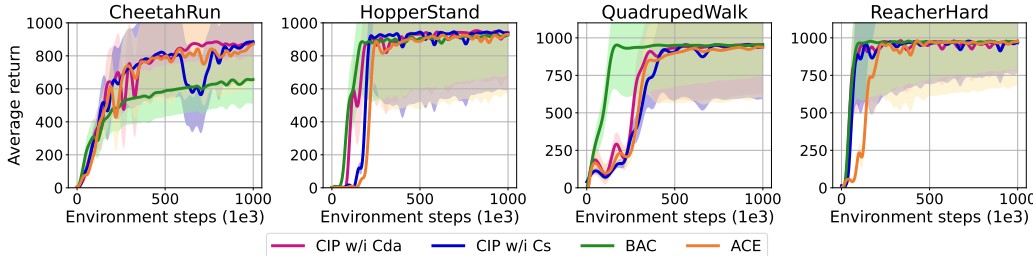

Figure 21: Experimental results in 4 locomotion tasks of DMControl environment.

also ensures that the agent can effectively navigate sparse reward scenarios by focusing on the most relevant causal information.

**Locomotion** We further conducted ablation experiments on locomotion tasks. The experimental results in the MuJoCo environment are shown in Figure 23, where it is evident that the performance of **CIP** without causal action empowerment declines significantly. Similarly, **CIP** without counterfactual data augmentation also exhibits reduced learning efficiency. Notably, in the 11 DMControl tasks, the decline in performance for **CIP** without causal action empowerment is particularly pronounced.

These experimental results further validate the effectiveness of our proposed method, which systematically analyzes the causal relationships between states, actions, and rewards. This analysis enables the execution of counterfactual data augmentation to avoid interference from irrelevant factors while prioritizing important state information. Subsequently, by leveraging the causal relationships between actions and rewards, we reweight actions to prioritize causally informed actions, thereby enhancing the agent's controllability and overall learning efficacy.

### D.3.3 HYPERPARAMETER ANALYSIS

We conduct a detailed analysis of the hyperparameters associated with the causal update interval ($I$) and sample size within the **CIP** framework. The experimental results for four distinct tasks are illustrated in Figure 25. Across all tasks, **CIP** demonstrates optimal performance with a causal update interval of $I = 2$ and a sample size of 10,000.

Our findings suggest that while a reduction in the causal update interval can lead to improved performance, it may also result in heightened computational costs. Additionally, we observe that higher update frequencies and increased sample sizes introduce greater instability, which significantly raises computational demands. This analysis underscores the importance of carefully balancing hyperparameter settings to optimize both performance and efficiency within the **CIP**.

Furthermore, we analyze the performance under different settings of the temperature factor $\alpha$ proposed in Eq. 9. The results across 3 tasks are shown in Figure 26. Our analysis reveals that **CIP** demonstrates robust performance across different values of $\alpha$ in manipulation tasks, while showing some instability in locomotion tasks when $\alpha$ is either too small or too large. Moreover, we observe that setting $\alpha$ to 0.2 yields optimal performance across all tasks, which motivated our choice of $\alpha = 0.2$ for all experiments.

Finally, we analyze the performance under different settings of the batch size and hidden size. The results across 3 tasks are shown in Figure 27. Our experimental results demonstrate that **CIP** exhibits robust performance across various parameter settings in coffee push and sparse hand insert tasks, while maintaining strong performance in hopper stand task. Based on these experimental results, we configure the hyperparameters as follows: for manipulation tasks, we set the batch size to 512 and hidden size to 1024, while for locomotion tasks, we use a batch size of 256 and hidden size of 256. All other hyperparameters remain constant across all tasks, as detailed in Table 3.

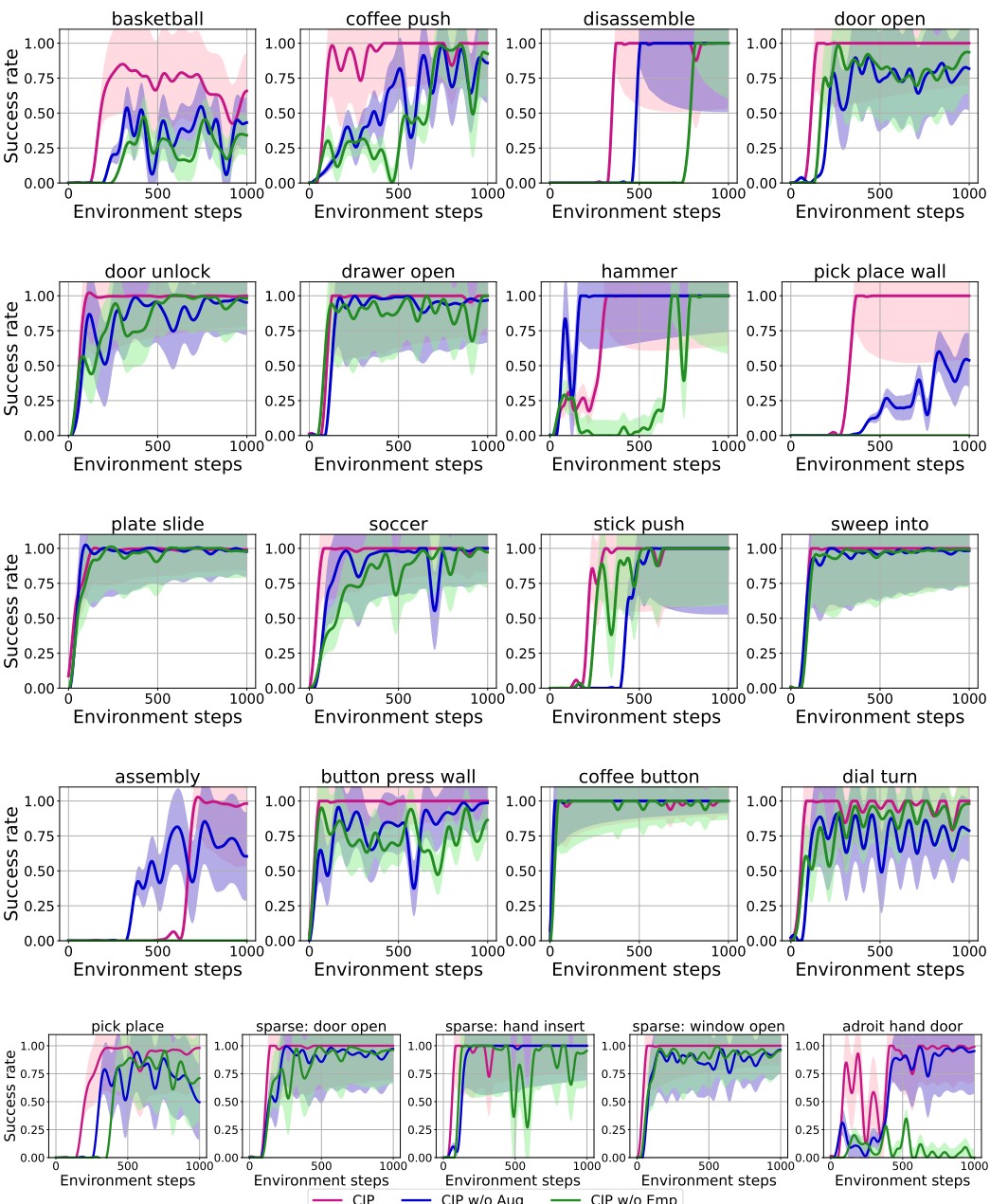

Figure 22: Ablation results across 21 manipulation skill learning tasks in Meta-World including sparse reward settings and adroit hand.

### D.3.4 COMPUTATION COST ANALYSIS

We analyze the computational cost of the proposed framework. The computation time for all methods across 36 tasks is shown in Figure 28. Our experimental results demonstrate that CIP achieves its performance improvements with minimal additional computational burden - specifically less than 10% increase compared to SAC, less than 5% increase compared to ACE, and actually requiring less computation time than BAC. All experiments were conducted on the same computing platform with the same computational resources detailed in Appendix F. These experimental results verify that our proposed method achieves performance improvements without incurring significant additional computational costs.

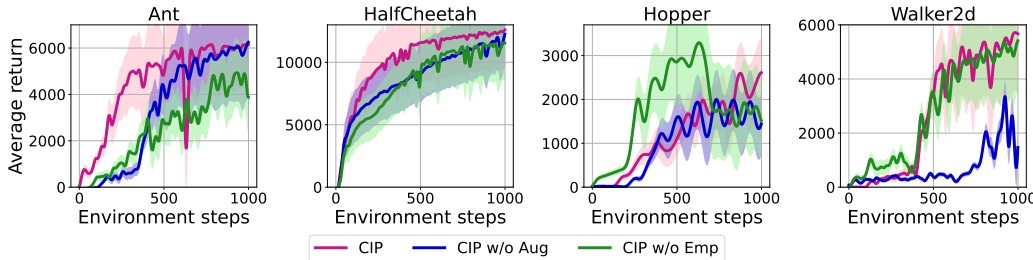

Figure 23: Ablation results across 4 locomotion tasks in MuJoCo environment.

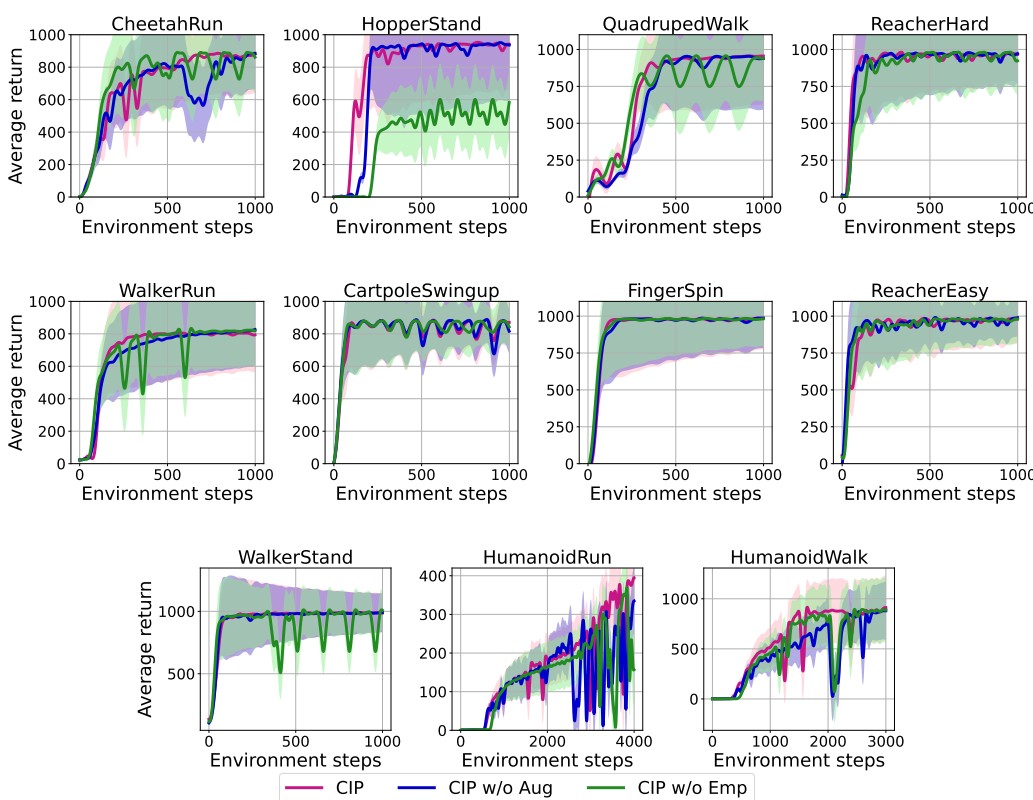

Figure 24: Ablation results across 11 locomotion tasks in DMControl environment.

### D.3.5 STATISTICAL PERFORMANCE ANALYSIS

To further validate the statistical significance of the performance, we select 3 statistical metrics (Agarwal et al., 2021) - IQM, Mean, and Median - for analysis across 8 locomotion tasks. The results are shown in Figure 29 and 30. Our findings indicate that **CIP** achieves notably superior performance across all tasks, with the sole exception of the ant task where it performs slightly below BAC.

### D.3.6 GENERALIZATION ANALYSIS

We conduct multi-task experiments in the Meta-World environment (Yu et al., 2020) to validate the generalizability. We establish MT1 and MT10 tasks for generalization validation:

**Multi-Task 1 (MT1)**: Learning one multi-task policy that generalizes to 5 tasks belonging to the same environment. MT1 uses single Meta-World environments, with the training "tasks" corresponding to

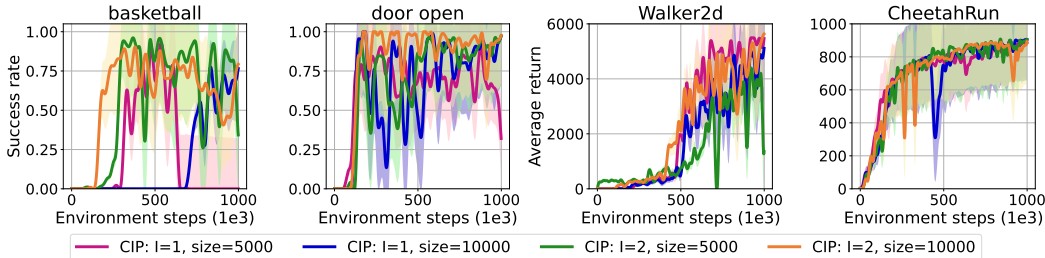

Figure 25: Hyperparameter study. Learning curves of **CIP** with different hyperparameter settings. The shaded regions are the standard deviation of each policy.

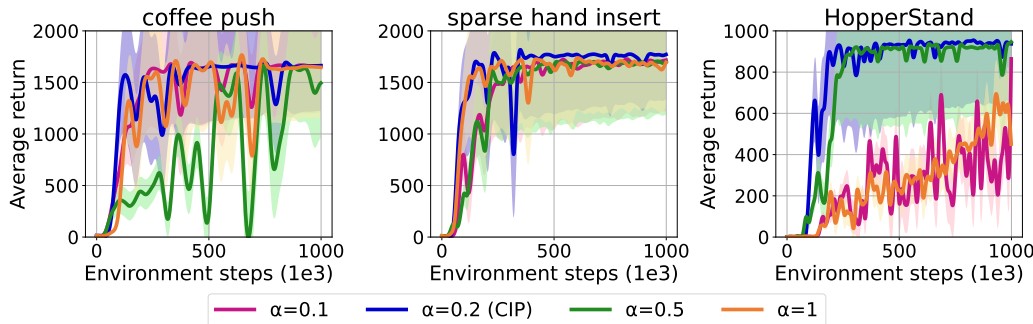

Figure 26: Hyperparameter analysis of temperature factor $\alpha$ across 3 task.

5 random initial object and goal positions. The goal positions are provided in the observation and are a fixed set, as to focus on the ability of algorithms in acquiring a distinct skill across multiple goals, rather than generalization and robustness.

**Multi-Task 10 (MT10)**: This task involves learning a single multi-task policy that generalizes to 50 tasks across 10 training environments, totaling 500 training tasks. A crucial step towards rapid adaptation to distinctly new tasks is the ability to train a single policy capable of solving multiple distinct training tasks. The multi-task evaluation in Meta-World tests the ability to learn multiple tasks simultaneously, without accounting for generalization to new tasks. The MT10 evaluation encompasses 10 environments: reach, push, pick and place, open door, open drawer, close drawer, press button top-down, insert peg side, open window, and close window.

We adapt our proposed **CIP** to multi-task learning by incorporating a one-hot task ID as input, comparing **MT−CIP** with MT-SAC. The results in Figure 31 show that **MT−CIP** outperforms MT-SAC in both MT1 (soccer) and MT10 tasks, achieving average success rates above $50\%$ and $40\%$ respectively. Notably, **MT−CIP** exhibits strong performance in specific MT10 tasks like drawer close and window open. The superior performance of **MT−CIP** stems from its effective learning of causal information, enabling robust task transfer across diverse domains. While these results are promising, future work will focus on causal state abstraction for enhanced generalization and sample efficiency. All experiments were conducted under the same hyperparameter settings, and the implementation will be made publicly available.

### D.3.7    CAUSAL DISCOVERY ANALYSIS

In **CIP**, we use the linear causal discovery method DirectLiNAM for causal structure learning. To explore alternative approaches, we compare it with two other causal discovery methods: score-based GES (Chickering, 2002) and constraint-based PC (Spirtes et al., 2001). The experimental results in Figure 32 across three tasks demonstrate that our chosen DirectLiNAM method exhibits superior performance compared to both alternatives. During experimentation, we also observe that both GES and PC methods incur significant computational overhead and frequently encounter memory constraints. In contrast, our proposed method **CIP**, which is fundamentally reward-guided, efficiently

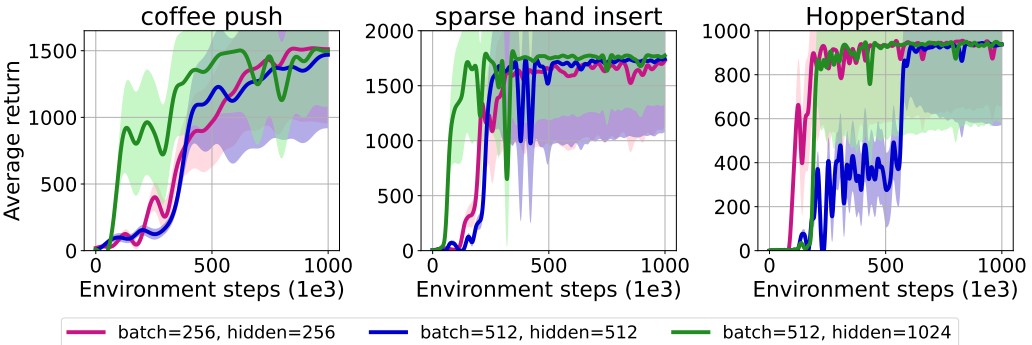

Figure 27: Hyperparameter analysis of batch size and hidden size across 3 task.

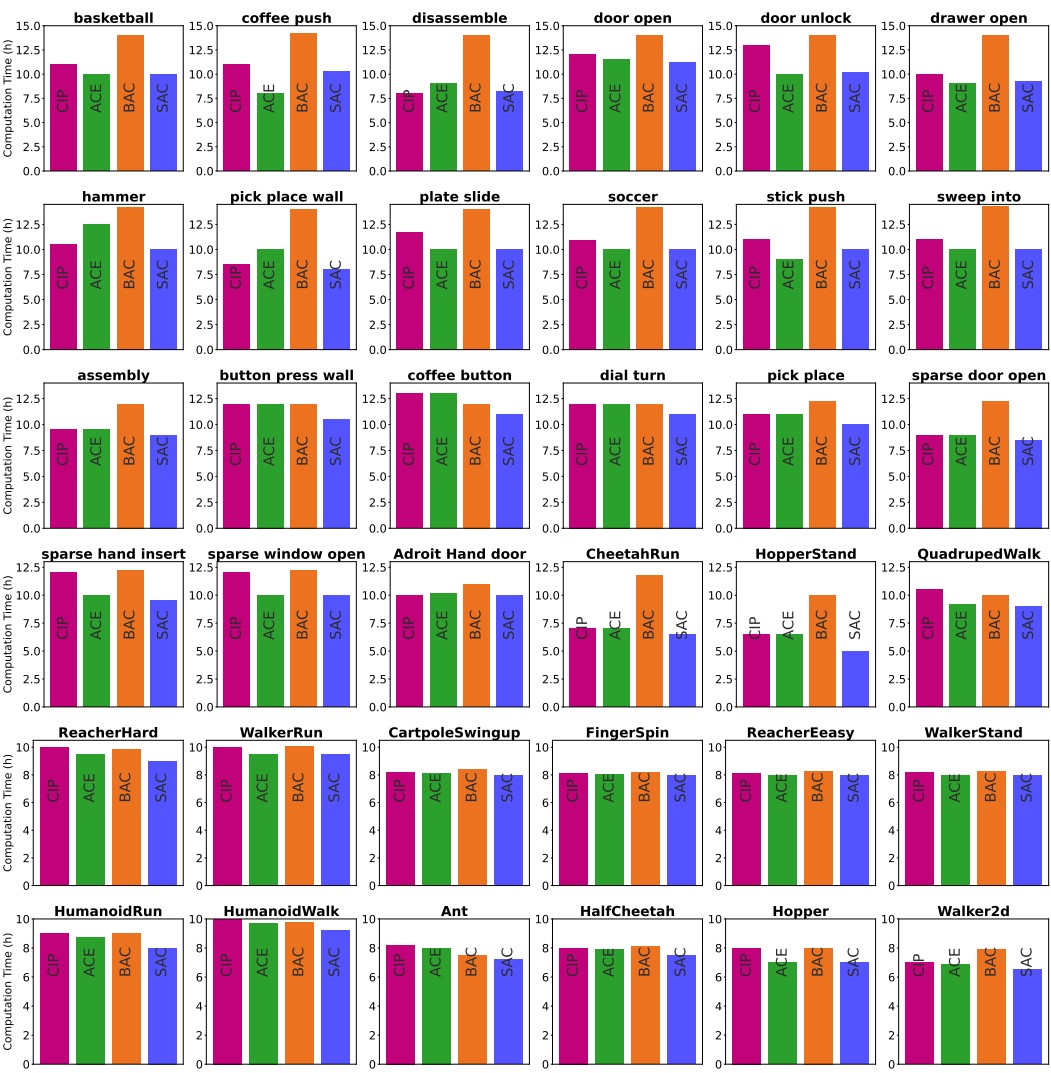

Figure 28: Computation time in 36 tasks.

discovers causal relationships between dimensional factors in states and actions with respect to rewards. This approach better aligns with the requirements of policy learning while maintaining minimal computational costs.

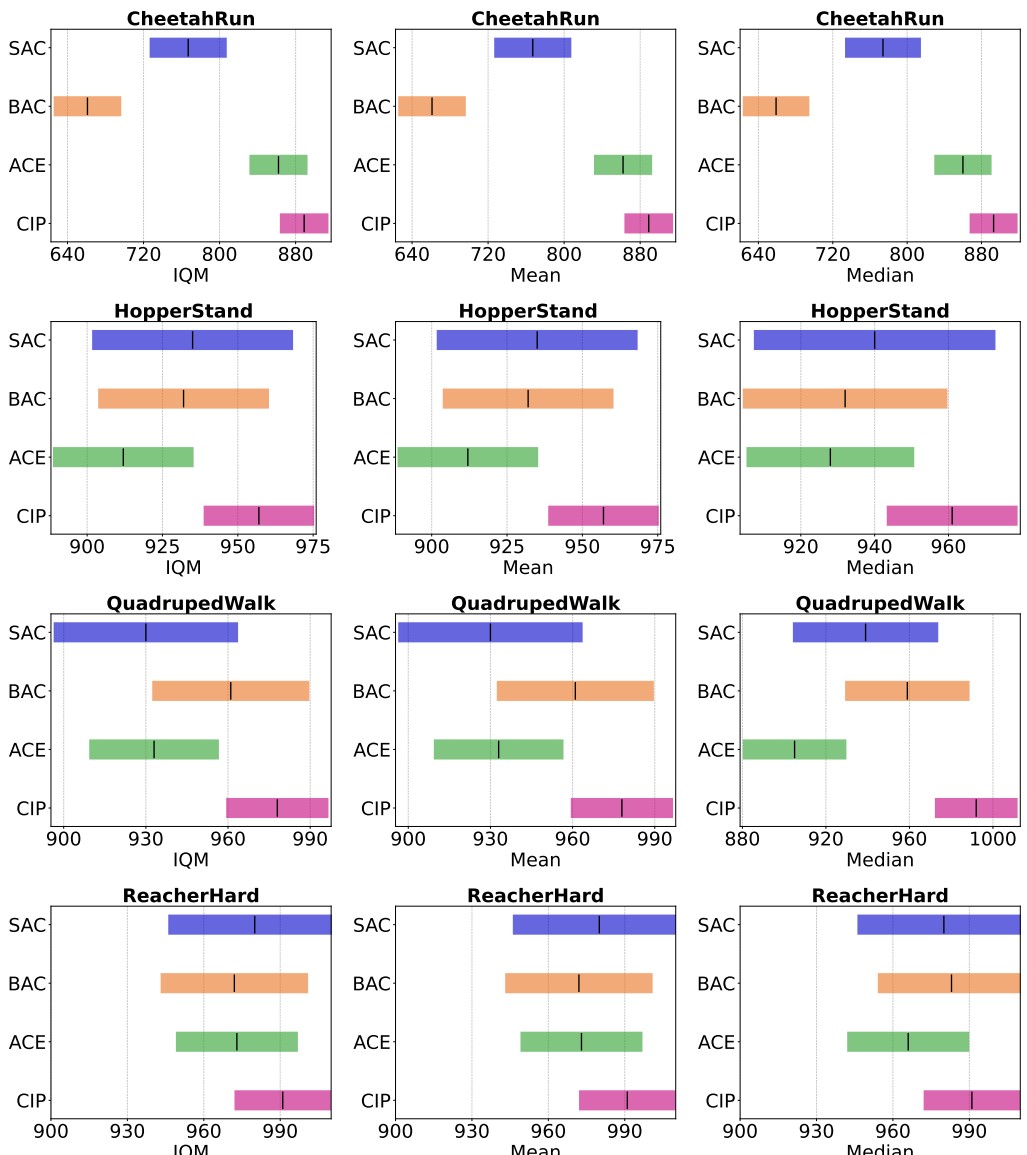

Figure 29: statistical metrics of IQM, Mean, and Median (higher values are better) on 4 DMControl tasks.

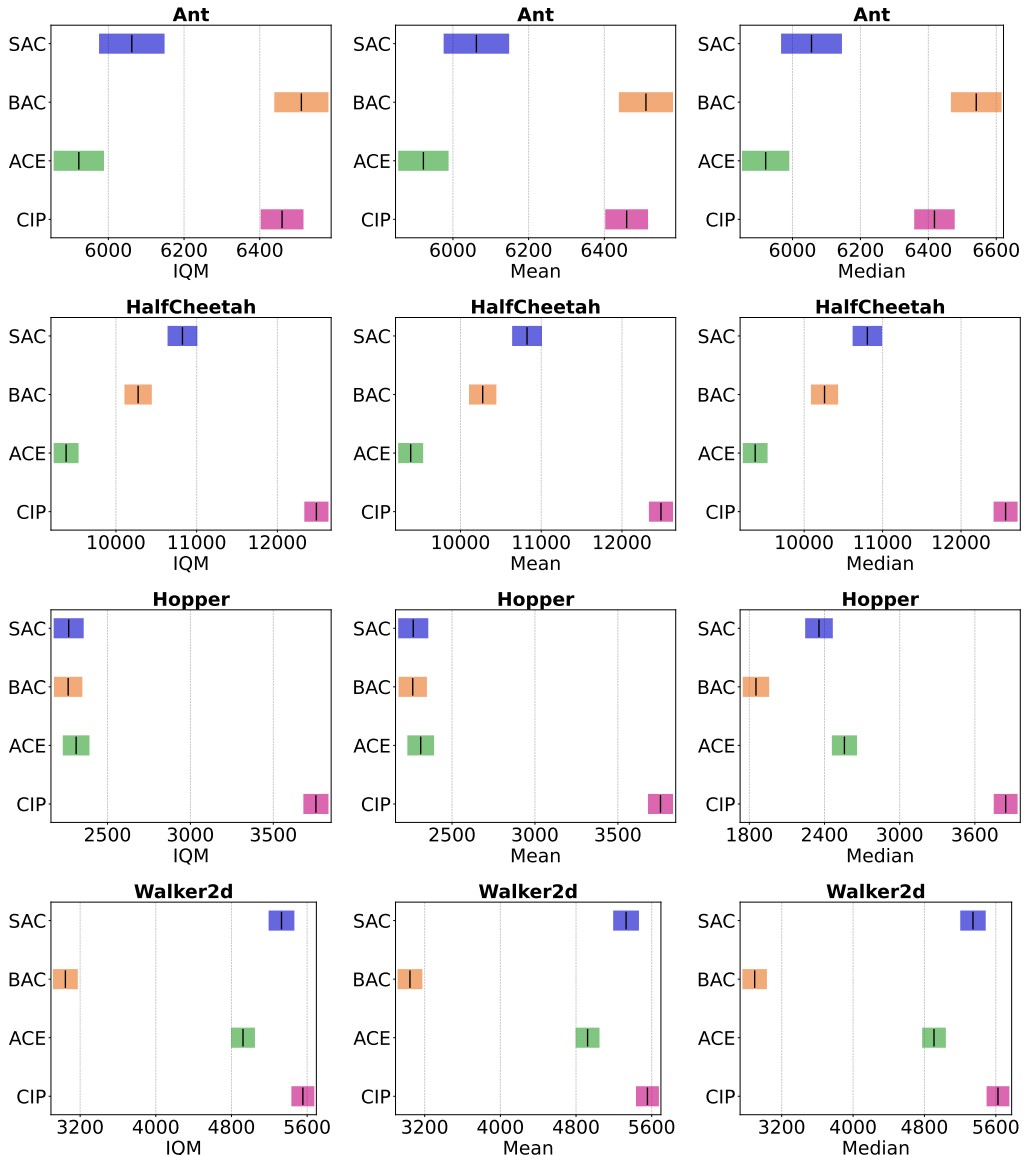

Figure 30: statistical metrics of IQM, Mean, and Median (higher values are better) on 4 MuJoCo tasks.

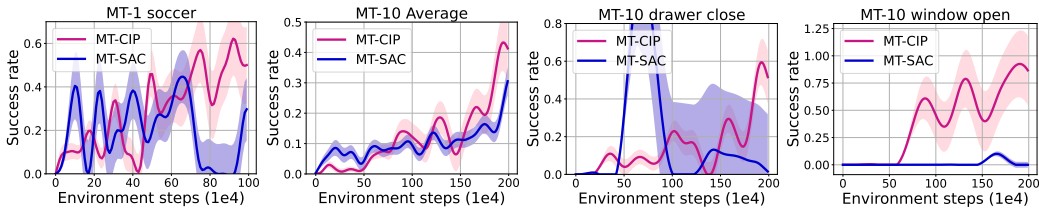

Figure 31: Generalization results in MT1 and MT10 tasks.

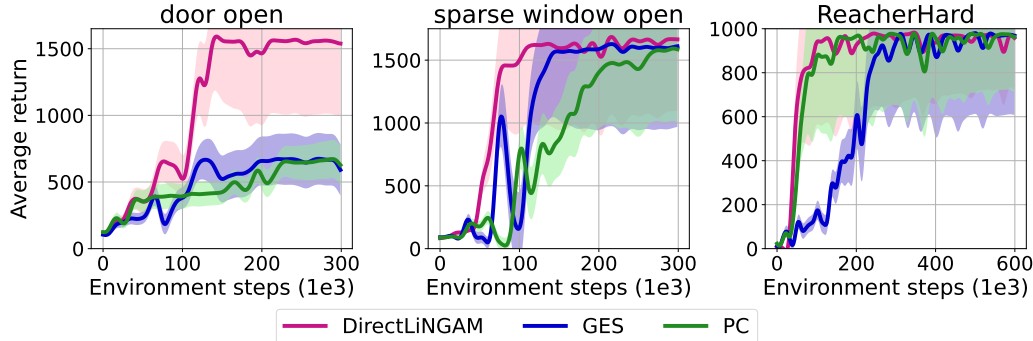

Figure 32: Compared performance with 2 different causal discovery methods across 3 task.

# E DETAILS ON THE PROPOSED FRAMEWORK

Algorithm 1 lists the full pipeline of `CIP` below.

# F EXPERIMENTAL PLATFORMS AND LICENSES

## F.1 EXPERIMENTAL PLATFORMS

All experiments of this approach are implemented on 2 Intel(R) Xeon(R) Gold 6430 and 2 NVIDIA Tesla A800 GPUs.

## F.2 LICENSES

In our code, we have utilized the following libraries, each covered by its respective license agreements:

- PyTorch (BSD 3-Clause "New" or "Revised" License)
- Numpy (BSD 3-Clause "New" or "Revised" License)
- Tensorflow (Apache License 2.0)
- Meta-World (MIT License)
- MuJoCo (Apache License 2.0)
- Deep Mind Control (Apache License 2.0)
- Adroit Hand (Creative Commons License 3.0)

---

**Algorithm 1** Causal information prioritization for efficient RL

---

**Input**: $Q$ network $Q_{\pi_c}$, policy network $\pi_c$, inverse dynamics model $\phi_c$ with $Q$ network $Q_{\phi_c}$, replay buffer $\mathcal{D}$, local causal buffer $\mathcal{D}_c$, causal update interval $I$, causal matrix $M^{a \to s}$ and $M^{a \to r}$.

    **for** each environment step $t$ **do**
        Collect data with $\pi_\theta$ from real environment
        Add to replay buffer $\mathcal{D}$ and local buffer $\mathcal{D}_c$
    **end for**

> **Step 1: Counterfactual data augmentation**
>
>     **if** every $I$ environment step **then**
>         Sample transitions $\mathcal{D}_s$ from local buffer $\mathcal{D}_c$
>         Learn causal mask matrix $M^{a \to r}$ with $\{(s, a, r, s')\}^{|\mathcal{D}_s|}$ for causal state prioritization
>         Compute uncontrollable set $\mathcal{U}_s$ followed by Eq. 4
>         Sample $(s, a, r, s') \in \mathcal{D}_s$
>         **for** $s^i \in \mathcal{U}_s$ **do**
>             Sample $(\hat{s}, \hat{a}, \hat{r}, \hat{s}') \sim \mathcal{D}_s$
>             **if** state $\hat{s}^i \in \mathcal{U}_{\hat{s}}$ **then**
>                 Construct a counterfactual transition $(\tilde{s}, \tilde{a}, \tilde{r}, \tilde{s}')$ by swapping $(s^i, s'^i)$ with $(\hat{s}^i, \hat{s}'^i)$
>                 Add $(\tilde{s}, \tilde{a}, \tilde{r}, \tilde{s}')$ to local buffer $\mathcal{D}_c$
>             **end if**
>         **end for**
>     **end if**

> **Step 2: Causal weighted matrix learning**
>
>     **if** every $I$ environment step **then**
>         Sample transitions $\mathcal{D}_a$ from local buffer $\mathcal{D}_c$
>         Learn causal weighted matrix $M^{a \to r}$ with $\{(s, a, r, s')\}^{|\mathcal{D}_a|}$ for causal action prioritization
>     **end if**

> **Step 3: Policy optimization with causal action empowerment**
>
>     **for** each gradient step **do**
>         Sample $N$ transitions $(s, a, r, s')$ from $\mathcal{D}$
>         Compute causal action empowerment followed by Eq. 8.
>         Calculate the target $Q_{\phi_c}$ value
>         Update $Q_{\phi_c}$ by $\min_{\phi_c}(\mathcal{T}Q_{\phi_c} - Q_{\phi_c})^2$
>         Update $\phi_c$ by $\max(Q_{\phi_c}(s, a))$
>         Calculate the target $Q_{\pi_c}$ value
>         Update $Q_{\pi_c}$ by $\min_{\pi_c}(\mathcal{T}_c Q_{\pi_c} - Q_{\pi_c})^2$
>         Update $\pi_c$ by $\max_c(Q_{\pi_c}(s, a) + \mathcal{E}_{\pi_c}(s))$
>     **end for**

