# OpenReview forum: "Causal Information Prioritization for Efficient Reinforcement Learning"
_ICLR.cc/2025/Conference — ICLR 2025 Poster_

### Official Review · Reviewer_hNGa · 2024-10-23

**Soundness:** 4
**Presentation:** 3
**Contribution:** 4
**Rating:** 8
**Confidence:** 5

**Summary:**

The paper proposes a novel reinforcement learning (RL) approach called Causal Information Prioritization (**CIP**), designed to address sample inefficiency by leveraging causal reasoning. CIP focuses on prioritizing causal relationships between states, actions, and rewards in reinforcement learning tasks, thus guiding the agent to focus on the most impactful behaviors. CIP improves sample efficiency by leveraging factored Markov Decision Processes (MDPs) to infer causal relationships and using counterfactual data augmentation. It integrates a causality-aware empowerment learning objective that enhances goal-directed action execution, leading to more efficient exploration. The approach is validated experimentally across 39 tasks in five diverse environments involving continuous control, demonstrating superior performance over existing RL techniques in terms of both sample efficiency and task success rates.

**Strengths:**

Integrating the causality-aware empowerment objective into the policy optimization objective alongside strong, extensive experimental results - demonstrating its ability to outperform several state-of-the-art RL methods, particularly in challenging environments with sparse rewards and high-dimensional action spaces - makes the paper a valuable contribution to the RL community.

**Originality**: The combination of causal reasoning with empowerment learning for reinforcement learning is a novel and creative approach.

**Quality**: The paper is well-researched and technically rigorous, with solid theoretical foundations and thorough experimental validation.

**Clarity**: The motivation and design of CIP are well-explained, with intuitive examples such as the robot-arm manipulation scenarios. The illustrations help make abstract concepts more accessible.

**Significance**: CIP addresses an important issue in reinforcement learning—sample inefficiency—by leveraging causal relationships, which has the potential for broad applicability in complex environments.

**Weaknesses:**

**Complexity of Empowerment Calculation**: The use of empowerment as part of the learning objective, though promising, involves additional computational complexity. The paper would benefit from a more in-depth discussion of how computational costs compare to the gains in sample efficiency, especially for tasks with very high-dimensional action spaces.

**Generalizability and Assumptions**: The reliance on the DirectLiNGAM method for causal discovery may limit generalizability in real-world environments where non-linear relationships are common. Exploring alternative causal discovery methods could enhance robustness.

**Empirical Comparisons**: While CIP is compared to standard baselines like SAC and ACE, further empirical comparisons to model-based RL approaches or hybrid methods would strengthen claims regarding sample efficiency and generalization capabilities.

**Questions:**

**Questions**

1. Could the authors elaborate on how CIP could be adapted or extended to discrete action spaces, where causal dependencies may be more rigid?

2. How does CIP handle potential **inaccuracies in causal discovery** during the learning process? Is there a mechanism in place to correct or adapt the causal models if they are misaligned with the environment dynamics?

3. **Causal Discovery**: Could the authors elaborate on the choice of DirectLiNGAM for causal discovery? Have other causal discovery techniques been tested, and if so, how do they compare in terms of performance and sample efficiency?

4. Assumptions of DirectLiNGAM: How sensitive is CIP to the assumptions required for the DirectLiNGAM method (e.g., linearity and Gaussian noise)? Would non-linear extensions of causal discovery alter CIP's effectiveness?

5. **Computational Overhead**: Could you provide a more detailed analysis of the computational costs associated with empowerment calculations compared to traditional entropy regularization approaches?

---

> ### Author Response · Authors · 2024-11-18
>
> We thank you for the encouraging and insightful comments. All of them are invaluable for further improving our manuscript. Please refer to our response below.
>
> > Q1: Computational Overhead: Could you provide a more detailed analysis of the computational costs associated with empowerment calculations compared to traditional entropy regularization approaches?
>
> **R1**: We appreciate your question regarding computational overhead. We analyze the computational cost of the proposed framework. The computation time for all methods including the traditional entropy regularization method SAC across 36 tasks is shown in Figure 28 of the revision. Our experimental results demonstrate that CIP achieves its performance improvements with minimal additional computational burden - specifically less than 10% increase compared to SAC, less than 5% increase compared to ACE, and actually requiring less computation time than BAC. For the detailed analysis, please refer to the Appendix D.3.4 of the revision.
>
> **Related Revised Sections: Figure 28, Appendix D.3.4**
>
> > Q2: Generalizability and Assumptions: The reliance on the DirectLiNGAM method for causal discovery may limit generalizability in real-world environments where non-linear relationships are common. Exploring alternative causal discovery methods could enhance robustness.
>
> **R2**: Thanks for your valuable suggestion. To explore alternative approaches, we compare DirectLiNGAM with two other causal discovery methods: score-based GES [1] and constraint-based PC [2]. We list the results of average return below. The learning curves are shown in Figure 32 of Appendix D.3.7 across three tasks in the revision. Our experimental results demonstrate that while GES and PC methods can be applied to certain tasks, they exhibit significantly lower learning efficiency. Based on these empirical results, we select DirectLiNGAM for causal discovery for simplicity and feasibility. For the detailed analysis, please refer to the Appendix D.3.7 of the revision.
> | Method       | door open | sparse window open | ReacherHard |
> |--------------|-----------|--------------------|-------------|
> | DirectLiNGAM | 1564      | 1655               | 991         |
> | GES          | 581       | 1600               | 953         |
> | PC           | 543       | 1580               | 963         |
>
> [1] Chickering, D. M. (2002). Optimal structure identification with greedy search. Journal of machine learning research, 3(Nov), 507-554.
>
> [2] Spirtes, P., Glymour, C. N., Scheines, R., & Heckerman, D. (2000). Causation, prediction, and search. MIT press.
>
> **Related Revised Sections: Figure 32, Appendix D.3.7**

---

> > ### Author Response · Authors · 2024-11-18
> >
> > > Q3: Empirical Comparisons: While CIP is compared to standard baselines like SAC and ACE, further empirical comparisons to model-based RL approaches or hybrid methods would strengthen claims regarding sample efficiency and generalization capabilities.
> >
> > **R3**: We appreciate your suggestion. Due to the absence of pre-trained dynamic models, CIP may inherently have lower learning efficiency compared to model-based methods, as is typical for model-free approaches. However, our comparisons with model-based approaches of MBPO [1], AutoMBPO [2], and SLBO [3] across three MuJoCo tasks (average return shown below and learning curves as reported in BAC [4]) demonstrate that CIP achieves superior performance and sample efficiency.
> > | Method   | Hopper      | Walker2d   | Ant        |
> > |----------|-------------|------------|------------|
> > | CIP      | 2846        | 5624       | 6418       |
> > | MBPO     | ~ 2500 | <5000 | < 6000 |
> > | AutoMBPO | ~ 2500 | < 4000 | < 6000 |
> > | SLBO     | ~ 0    | < 3000 | < 2000 |
> >
> > Furthermore, regarding generalization capabilities, we have conducted comprehensive multi-task experiments, with detailed results presented in Appendix D.3.6. We use MT1 (soccer) and MT10 tasks designed in MetaWorld [5] for generalization validation. CIP outperforms SAC in both tasks, achieving average success rates above 50% and 40% respectively. The results demonstrate CIP's good performance in multi-task settings, enabling robust knowledge transfer across diverse domains.
> >
> > [1] Janner, Michael, et al. "When to trust your model: Model-based policy optimization." Advances in neural information processing systems 32 (2019).
> >
> > [2] Lai, Hang, et al. "On effective scheduling of model-based reinforcement learning." Advances in Neural Information Processing Systems 34 (2021): 3694-3705.
> >
> > [3] Luo, Yuping, et al. "Algorithmic Framework for Model-based Deep Reinforcement Learning with Theoretical Guarantees." International Conference on Learning Representations. 2019
> >
> > [4] Ji, Tianying, et al. "Seizing Serendipity: Exploiting the Value of Past Success in Off-Policy Actor-Critic." Forty-first International Conference on Machine Learning. 2024
> >
> > [5] Yu, Tianhe, et al. "Meta-world: A benchmark and evaluation for multi-task and meta reinforcement learning." Conference on robot learning. PMLR, 2020.
> >
> > **Related Revised Sections: Appendix D.3.6**
> >
> > > Q4: Could the authors elaborate on how CIP could be adapted or extended to discrete action spaces, where causal dependencies may be more rigid?
> >
> > **R4**: Thank you for this insightful question. Yes, CIP can be naturally adapted to discrete domains by directly analyzing the causal relationships between discrete action factors and rewards. This is generally flexible to be adapted to those environments. In the current framework, we use SAC as our policy learning module, so we do not test them on discrete environments. To make it more complete, we are running one additional experiment on the Cartpole environment, where the action space is discrete. We will post the results here when the results are ready.
> >
> > > Q5: How does CIP handle potential inaccuracies in causal discovery during the learning process? Is there a mechanism in place to correct or adapt the causal models if they are misaligned with the environment dynamics?
> >
> > **R5**: Thank you for this insightful point. Yes, our approach relies on an accurate causal model to guide the agent's decisions. In the RL environment we used, unlike standard causal discovery benchmarks, we do not have access to the ground-truth causal structures. As such, we have not conducted a rigorous analysis of causal structure accuracy in this work, but we plan to address this in future research by creating some benchmarks.
> > Regarding potential errors in the learned causal model, the empowerment optimization term can implicitly help refine the causal structure. This term encourages the selection of actions that lead to predictable outcomes or state dimensions, given the current and subsequent states as determined by the causal structure. By optimizing this objective, the model is guided toward uncovering the true dynamics structure, effectively mitigating errors in the learned causal relationships.

---

> ### Author Response · Authors · 2024-11-20
> **Update on 20/Nov: Results on experiments with discrete action space**
>
> Thank you once again for your insightful review. We have finished the experiments with discrete action space.
>
> > **Q4**: Could the authors elaborate on how CIP could be adapted or extended to discrete action spaces, where causal dependencies may be more rigid?
>
> **Additional-R4**: We evaluate CIP in the Cartpole environment of discrete action spaces. The results of average return are shown below and the learning curves are shown in the revised Figure 6. These results further demonstrate the superior sample efficiency and performance of CIP in discrete action spaces.
> |         | Cartpole |
> |---------|----------|
> | CIP     | 186     |
> | IFactor | 183     |
>
> **Related Revised Sections: Figure 6, Section 5.2**

---

> > ### Author Response · Authors · 2024-11-25
> >
> > We would like to thank you once again for your encouraging and constructive review. We have provided responses above, including additional evaluations and discussions on computational cost analysis (**R1**), more causal discovery methods (**R2**), comparisons with MBRL baselines (**R3**), experiments with discrete action spaces (**R4**), and the effect of inaccurate causal discovery during the model learning stage (**R5**). The points you raised were invaluable for improving this work, and we have incorporated the new results and discussions into the revised main paper and appendix.
> >
> > If you have any further questions about the paper or this rebuttal, please feel free to reach out. We would be more than happy to address or discuss them further. Thank you again for your time and effort!

---

> > > ### Comment · Reviewer_hNGa · 2024-11-25
> > > **Re: Author Responses**
> > >
> > > Dear Authors,
> > > Thank you for your detailed answers and revised manuscript  - my questions have been effectively answered. I've increased my confidence of my (existing) high-rating of your work. Best of luck.

---

> ### Comment · Area_Chair_FDKw · 2024-11-25
> **Rebuttal feedbacks**
>
> Dear  Reviewer hNGa,
> Could you please read the authors' rebuttal and give them feedback at your earliest convenience? Thanks.
> AC

---

> ### Author Response · Authors · 2024-11-26
> **Thank you!**
>
> Thank you for your encouraging comments and we are pleased that we could effectively address your concerns. Your insights have been invaluable to improving our work. We sincerely appreciate your time and effort!

---

### Official Review · Reviewer_kTcs · 2024-11-03

**Soundness:** 4
**Presentation:** 3
**Contribution:** 3
**Rating:** 6
**Confidence:** 4

**Summary:**

This paper introduces CIP, the causal information prioritization method that leverages causal information in reinforcement learning to improve learning efficiency. Comprehensive experiments are conducted to verify the proposed method.

**Strengths:**

The paper is well-organized. The presentation is mostly clear. The experiment section is comprehensive.

**Weaknesses:**

1. Several related works are missed:

[1] Sun, Yuewen, et al. "ACAMDA: Improving Data Efficiency in Reinforcement Learning Through Guided Counterfactual Data Augmentation." Proceedings of the AAAI Conference on Artificial Intelligence. Vol. 38. No. 14. 2024.

[2] Sun, Hao, and Taiyi Wang. "Toward Causal-Aware RL: State-Wise Action-Refined Temporal Difference." arXiv preprint arXiv:2201.00354 (2022).

2. The limitation section in the current paper is not really a discussion of the limitations of the proposed method.

**Questions:**

Will the proposed method be suitable for offline RL or has it to be applied with online explorations?

Can the authors provide more analysis on the pitfalls of the proposed method? What is the trade-off of the proposed method, and under what cases would this method face challenge? e.g. computational burden, can not handle highly stochastic environments?

I appreciate the authors' efforts in their large-scale experiments, yet I have concerns about the statistical significance of the performance. On some of the tasks, the performance seems to have large variances. I wonder what would be the recommended hyper-parameter setups from the authors, and is the proposed method stable/robust to different hyper-parameter settings?

---

> ### Author Response · Authors · 2024-11-18
>
> We thank the reviewer for the insightful and encouraging comments. Please see our response and discussions as follows.
>
> > Q1: Several related works are missed.
>
> **R1**: Thank you for the pointers. We have discussed these works in the related work section (Sec 2.1).
>
> **Related Revised Sections: Sec 2.1**
>
> > Q2: The limitation section in the current paper is not really a discussion of the limitations of the proposed method.
>
> **R2**: Thanks for the advice, here we further discuss the limitations which we have revised in the revision (Sec 6). The current limitations of our work are twofold. First, CIP has not yet been extended to complex scenarios, such as real-world 3D robotics tasks. Potential approaches to address this limitation include leveraging object-centric models [1], 3D perception models [2], and robotic foundation models [3, 4] to construct essential variables for causal world modeling (A more detailed discussion is in Appendix C.1). Second, CIP does not adequately consider non-stationarity and heterogeneity, which might exist in real-world environments. Future work could integrate method designed to handle non-stationarity and heterogeneity in our causal discovery module, such as [5].
>
> [1] Wu, Ziyi, et al. "Slotdiffusion: Object-centric generative modeling with diffusion models." Advances in Neural Information Processing Systems 36 (2023): 50932-50958.
>
> [2] Wang, Chenxi, et al. "RISE: 3D Perception Makes Real-World Robot Imitation Simple and Effective." ICRA 2024 Workshop on 3D Visual Representations for Robot Manipulation.
>
> [3] Team, Octo Model, et al. "Octo: An open-source generalist robot policy." arXiv preprint arXiv:2405.12213 (2024).
>
> [4] Firoozi, Roya, et al. "Foundation models in robotics: Applications, challenges, and the future." The International Journal of Robotics Research (2023): 02783649241281508.
>
> [5] Huang, Biwei, et al. "Causal discovery from heterogeneous/nonstationary data." Journal of Machine Learning Research 21.89 (2020): 1-53.
>
> **Related Revised Sections: Sec 6, Appendix C.1**
>
> > Q3: Will the proposed method be suitable for offline RL or has it to be applied with online explorations?
>
> **R3**: Great point! In the current work, our focus is on using action empowerment for improved online exploration. However, in future work, we plan to explore the potential of action empowerment in offline RL. Here are a few ideas we are considering: (1) Learn the causal structure from offline data, and then use the learned structure to perform causality-guided data augmentation, generating counterfactual data that aligns with similar empowerment targets (as described in Section 4.1); (2) Since offline RL faces algorithmic challenges due to function approximation errors caused by out-of-distribution (OOD) data points [1, 2], we could combine empowerment regularization with ensembling or uncertainty quantification techniques, as applied in SAC, and use the same target as in Section 4.2. As these are basically orthogonal to the major contribution of this work,  we plan to explore these directions in future work.
>
> [1] Yu T, Thomas G, Yu L, et al. Mopo: Model-based offline policy optimization[J]. Advances in Neural Information Processing Systems, 2020, 33: 14129-14142.
>
> [2] Kumar A, Zhou A, Tucker G, et al. Conservative q-learning for offline reinforcement learning[J]. Advances in Neural Information Processing Systems, 2020, 33: 1179-1191.

---

> ### Author Response · Authors · 2024-11-18
>
> > Q4: Can the authors provide more analysis on the pitfalls of the proposed method? What is the trade-off of the proposed method, and under what cases would this method face challenge? e.g. computational burden, can not handle highly stochastic environments?
>
> **R4**: Thanks for your suggestion. CIP may face challenges in real-world robotics scenarios with high-dimensional, complex, and noisy observations, where the DirectLiNGAM causal discovery method might struggle to effectively capture causal relationships between states, actions, and rewards.  However, as we describe in the updated limitations part (Sec 6) and Appendix C.1, for these highly complex, real-world environments, we can leverage the current state-of-art 3D world model, object-centric models as encoders to help us learn essential abstraction variables from the observation, and then we can still employ the current framework. As these are very practical applications of robotics with heavy engineering extensions, we will leave these as future work.
>
> Regarding computational burden, we analyze the computational cost of the proposed framework. The computation time for all methods across 36 tasks is shown in Figure 28 of the revision. Our experimental results demonstrate that CIP achieves its performance improvements with minimal additional computational burden - specifically less than 10% increase compared to SAC, less than 5% increase compared to ACE, and actually requiring less computation time than BAC. For the detailed analysis, please refer to the Appendix D.3.4 of the revision.
>
> **Related Revised Sections: Figure 28, Sec 6, Appendix C.1, Appendix D.3.4**
>
> > Q5: I appreciate the authors' efforts in their large-scale experiments, yet I have concerns about the statistical significance of the performance. On some of the tasks, the performance seems to have large variances. I wonder what would be the recommended hyper-parameter setups from the authors, and is the proposed method stable/robust to different hyper-parameter settings？
>
> **R5**: Thank you for this valuable feedback. To address your concerns about statistical significance, we have conducted comprehensive statistical analyses using four distinct metrics across eight tasks, with detailed results presented in Table 1 and Appendix D.3.5 of the revision. T-test analyses demonstrate that CIP achieves statistically significant improvements in 5 out of 8 tasks (Table 1). Additional statistical analyses using IQM, Mean, and Median metrics reveal that CIP outperforms baselines in 7 out of 8 tasks (Appendix D.3.5).
> Furthermore, to evaluate the robustness of our method, we have performed extensive hyperparameter sensitivity studies, examining performance across different settings of the temperature factor, batch size, and hidden size. Experimental results below of average return and learning curves in Figure 26 and 27 of the revision, demonstrating that CIP exhibits robust performance across various parameter settings in manipulation tasks, while maintaining strong performance in locomotion tasks. Notably, all other hyperparameters remain constant across tasks (Table 3), demonstrating the method's feasibility. These detailed analyses can be found in Appendix D.3.3 of the revision.
> | α   | coffee push | sparse hand insert | HopperStand |
> |-----|-------------|--------------------|-------------|
> | 0.1 | 1631        | 1760               | 931         |
> | 0.2 | 1661        | 1783               | 934         |
> | 0.5 | 1623        | 1714               | 947         |
> | 1   | 1613        | 1735               | 760         |
>
>
> | batch size | hidden size | coffee push | sparse hand insert | HopperStand |
> |------------|-------------|-------------|--------------------|-------------|
> | 256        | 256         | 1481        | 1735               | 953         |
> | 512        | 512         | 1661        | 1743               | 938         |
> | 512        | 1024        | 1661        | 1776               | 950         |
>
> **Related Revised Sections: Figure 26, Figure 27, Table 1, Table 3, Appendix D.3.3, Appendix D.3.5**

---

> > ### Author Response · Authors · 2024-11-25
> >
> > Thank you for your insightful comments and review. We have provided responses above, focusing on discussions of related work, limitations, potential pitfalls, and future directions (**R1**, **R2**, **R4**), as well as the discussion on the potential of our approach to offline RL (**R3**). Additionally, we conducted further evaluations, including statistical significance testing of performance and hyperparameter ablations (**R5**). All details have been incorporated into the revised paper, with pointers given in the responses above.
> >
> > If you have any further comments or questions, please feel free to let us know. We would be more than happy to address or discuss them before this rebuttal session closes. Thank you once again for your valuable contribution to reviewing this work!

---

> > > ### Comment · Reviewer_kTcs · 2024-11-25
> > > **Thanks for the detailed response!**
> > >
> > > I appreciate the authors' detailed response and their effort in improving the paper.
> > >
> > > I do not have further concerns and I've increased my soundness rating.

---

> > > > ### Author Response · Authors · 2024-11-25
> > > > **Thank you!**
> > > >
> > > > We are glad that we were able to address your concerns. And thank you for raising the soundness score—we sincerely appreciate your time and effort!

---

### Official Review · Reviewer_wgJu · 2024-11-07

**Soundness:** 2
**Presentation:** 3
**Contribution:** 2
**Rating:** 5
**Confidence:** 3

**Summary:**

This paper introduces Causal Information Prioritization (CIP) to address sample inefficiency in reinforcement learning by leveraging causal relationships between states, actions, and rewards. CIP uses factored Markov Decision Processes (MDPs) to model these causal relationships and employs techniques like counterfactual data augmentation to improve sample efficiency. The framework is tested across various tasks, including manipulation and locomotion, to showcase its effectiveness in different environments.

**Strengths:**

- The paper presents an interesting approach by integrating causal reasoning with reinforcement learning (RL), which could potentially lead to better sample efficiency by filtering out irrelevant features.
- The focus on causality and counterfactual augmentation aligns with a growing area in RL, aiming to make models more effective with fewer interactions by prioritizing meaningful features.
- The experiments cover diverse environments and tasks, which is valuable for assessing the generalizability of the approach.

**Weaknesses:**

- The paper seems to need more references and discussion around object-centric MDPs and related works in object-centric world modeling and MDP homomorphism, which are relevant areas given the nature of CIP.
- The proposed framework appears to require considerable task-specific engineering, as it doesn’t seem to inherently learn the causal structure but rather relies on manually provided task structures. This may limit its scalability and adaptability to new tasks without re-engineering.
- Empirical results, especially in Table 1, show limited improvement over baselines like ACE and BAC. The gains aren’t always significant, especially given the complexity and additional engineering that CIP requires. The benefits may not justify the added complexity in some cases.
- Some results in Table 1 indicate the CIP approach performs on par with or even below baselines for certain tasks, suggesting that the improvements might not generalize strongly across all evaluated environments.

**Questions:**

- How does CIP address the need for task-specific causal structure input? Is it possible to reduce the engineering effort required for this, or is the approach dependent on manual structure provision?
- Were there specific reasons for the chosen baselines (ACE, BAC)? Would other baselines, particularly those that handle causality differently, add more context to the results?
- What are the plans for integrating 3D world understanding or extending CIP to work more naturally with object-centric or high-dimensional MDPs without predefined causal structures?

---

> ### Author Response · Authors · 2024-11-18
>
> We thank the reviewer for the insightful and useful feedback, please see the following for our response.
>
> > Q1: The paper seems to need more references and discussion around object-centric MDPs and related works in object-centric world modeling and MDP homomorphism, which are relevant areas given the nature of CIP.
>
> **R1**: Thank you for the suggestion. We have expanded the related works section (Section 2.3) to include discussions on object-centric learning, object-centric RL, object-oriented RL, and MDP homomorphisms. Specifically, we now address both empirical and theoretical works in object-centric representation learning, including their use in compositional generalization, world models, and image/video/scene generation. Additionally, we cover related works that utilize object-centric learning for RL, such as employing object-centric world models for modeling dynamics, identifying and learning object-centric policies, and using object or interaction representations for intrinsic rewards and curiosity. Furthermore, we discuss object-oriented MDPs and MDP homomorphism.
>
> Basically, object-centric RL and the causal structure learned in our framework share the similarity of both using factored MDPs to model the environment, where ours focus on the raw states but object-centric learns the object-wise factors. We agree that object-centric models can be highly valuable for extending CIP in specific applications (such as real-world robotic manipulation with numerous objects), as they provide useful abstractions of observations—such as object attributes, states, and relationships. However, our current framework, which focuses on learning causal models and empowerment, is orthogonal to these works. We believe combining object-centric representations (as strong representation encoders and dynamics models) with causal representation learning is an exciting direction for future work. We have provide a detailed discussion in revised Appendix C.1, listing three potential directions for future work: (1) Use object-centric representation as input for causal structure learning; (2) Learning more compact object-centric representation with casual discovery; (3) Use object-aware 3D models as encoders for real-world robotics tasks. While these directions are promising and could advance the applicability of our framework in certain domains, they are outside the primary focus of this work. We plan to explore these ideas as part of future work.
>
>
> **Related Revised Sections: Sec 2.3, Appendix C.1**
>
> >  Q2: The proposed framework appears to require considerable task-specific engineering, as it doesn’t seem to inherently learn the causal structure but rather relies on manually provided task structures. This may limit its scalability and adaptability to new tasks without re-engineering.
>
> **R2**: We would like to clarify that the causal structures across different tasks are automatically learned using causal discovery methods from observational data, rather than being manually engineered. Though different tasks exhibit distinct causal structures, our approach leverages a unified workflow to address this variability.
> Additionally, we have incorporated more causal discovery methods in this rebuttal. In the original paper, we employed DirectLiNGAM due to its simplicity and flexibility. To provide a more comprehensive evaluation, we have now included results using additional methods, specifically score-based GES [1] and constraint-based PC [2]. We list the results of average return below. The learning curves across three tasks are shown in Figure 32 of Appendix D.3.7 in the revision. Our experimental analysis demonstrates that while GES and PC methods can be applied to certain tasks, they exhibit lower learning efficiency compared to DirectLiNGAM.
> | Method       | door open | sparse window open | ReacherHard |
> |--------------|-----------|--------------------|-------------|
> | DirectLiNGAM | 1564      | 1655               | 991         |
> | GES          | 581       | 1600               | 953         |
> | PC           | 543       | 1580               | 963         |
>
> [1] Chickering, D. M. (2002). Optimal structure identification with greedy search. Journal of machine learning research, 3(Nov), 507-554.
>
> [2] Spirtes, P., Glymour, C. N., Scheines, R., & Heckerman, D. (2000). Causation, prediction, and search. MIT press.
>
> **Related Revised Sections: Appendix D.3.7**

---

> ### Author Response · Authors · 2024-11-18
>
> > Q3: Results in Table 1 indicate the CIP approach performs on par with or even below baselines for certain tasks, suggesting that the improvements might not generalize strongly across all evaluated environments.
>
> **R3**: Thank you for pointing this out. To validate our performance improvements in Table 1, we have conducted statistical analysis by pair-wise t-test on our method with baselines all 8 tasks, demonstrating that CIP achieves statistically significant improvements in 5 out of 8 tasks, with detailed results in Table 1 of the revision. Additional statistical analysis using IQM, Mean, and Median metrics [1] show that CIP outperforms baselines in 7 out of 8 tasks, with detailed results available in Figure 29 and 30 in the Appendix D.3.5 of the revision.
> Furthermore, we also evaluate the generalization capabilities of multi-task learning (though our framework is not designed for multi-task RL specifically), we have performed extensive multi-task experiments, shown in Appendix D.3.6. We use MT1 (soccer) and MT10 tasks designed in MetaWorld [2] for generalization validation. CIP outperforms SAC in both tasks, achieving average success rates above 50% and 40% respectively.
>
> [1] Agarwal, Rishabh, et al. "Deep reinforcement learning at the edge of the statistical precipice." Advances in neural information processing systems 34 (2021): 29304-29320.
>
> [2] Yu, Tianhe, et al. "Meta-world: A benchmark and evaluation for multi-task and meta reinforcement learning." Conference on robot learning. PMLR, 2020.
>
> **Related Revised Sections: Table 1, Figure 29, Figure 30, Appendix D.3.5, Appendix D.3.6**
>
> > Q4: Were there specific reasons for the chosen baselines (ACE, BAC)? Would other baselines, particularly those that handle causality differently, add more context to the results?
>
> **R4**: We chose ACE and BAC primarily because these two methods represent the current state-of-the-art sample-efficient RL in continuous control. Specifically, ACE learns the causal structure between actions and rewards, identifying the primitive actions that contribute to efficiency. And BAC balances sufficient exploitation of past successes with exploration optimism. Thus, we believe these methods provide strong and relevant baselines for comparison.
>
> For additional causal discovery approaches, particularly in the context of causal discovery and considering the reviewer’s suggestion, we select score-based GES and constraint-based PC methods for comparison. Please refer to the response to question 2 (Q2), with detailed analysis in Appendix D.3.7 of the revision.
>
> **Related Revised Sections: Appendix D.3.7**
>
> > Q5: What are the plans for integrating 3D world understanding or extending CIP to work more naturally with object-centric or high-dimensional MDPs without predefined causal structures?
>
> **R5**: Thank you for pointing out this discussion. Yes, we think integrating 3D representation learning with causal world models in this work could be a promising future direction. Specifically, we can use object-centric representation learning [1-3] or 3D robotics perception models [4] and the intersections like object-centric Gaussian Splatting [5] to give us the 3D object-centric states and representation, and then we can use the object-centric states as variables in this framework and the remaining steps in this paper can be reused. In that sense, the causal structures are among the 3D objects.  We will leave this for future work.  We also provide a detailed discussion in revised Appendix C.1.
>
> [1] Locatello, Francesco, et al. "Object-centric learning with slot attention." Advances in neural information processing systems 33 (2020): 11525-11538.
>
> [2] Wu, Ziyi, et al. "Slotdiffusion: Object-centric generative modeling with diffusion models." Advances in Neural Information Processing Systems 36 (2023): 50932-50958.
>
> [3] Aydemir, Görkay, Weidi Xie, and Fatma Guney. "Self-supervised object-centric learning for videos." Advances in Neural Information Processing Systems 36 (2023): 32879-32899.
>
> [4] Wang, Chenxi, et al. "RISE: 3D Perception Makes Real-World Robot Imitation Simple and Effective." ICRA 2024 Workshop on 3D Visual Representations for Robot Manipulation.
>
> [5] Li, Yulong, and Deepak Pathak. "Object-Aware Gaussian Splatting for Robotic Manipulation." ICRA 2024 Workshop on 3D Visual Representations for Robot Manipulation.
>
> **Related Revised Sections: Appendix C.1**

---

> > ### Author Response · Authors · 2024-11-25
> >
> > Thank you for your constructive comments. We have provided responses addressing key points, including discussions on related works and future directions (**R1**, **R5**), additional evaluations with more causal discovery methods (**R2**), significance testing of the performance (**R3**), and the rationale behind the baseline selections (**R4**). Revisions based on these points have been made in the paper, with pointers to the revised sections provided in the response above.
> >
> > As the discussion deadline approaches, please feel free to let us know if you have any additional comments or questions. We would be more than happy to address and discuss them. Thank you again for your time and effort!

---

> ### Comment · Area_Chair_FDKw · 2024-11-25
> **Rebuttal feedbacks**
>
> Dear Reviewer wgJu,
> Could you please read the authors' rebuttal and give them feedback at your earliest convenience? Thanks.
> AC

---

> > ### Author Response · Authors · 2024-11-27
> > **Thank you for the insightful comments, we are happy to take further questions if any**
> >
> > Dear Reviewer wgJu,
> >
> > Thank you once again for your thoughtful review and valuable feedback! As the revision deadline approaches **in a few hours**, we wanted to check if you have any additional comments or concerns on our rebuttal. We would be happy to discuss and address them in the upcoming revision (if shared before the deadline). If not, we are also still more than happy to address any further questions or concerns in this rebuttal thread until the final discussion deadline. Please feel free to let us know at your convenience.
> >
> > Best regards,
> >
> > Submission 1433 Authors

---

### Author Response · Authors · 2024-11-22
**General response**

We sincerely appreciate the thoughtful feedback and encouraging comments from all reviewers. Your insights have been invaluable in improving the clarity of our work and identifying key points for deeper analysis and future works. We have addressed each point in the individual rebuttals and also revised our main paper and appendix accordingly, **with changes marked in red for clarity.**

We are grateful for the reviewers' recognition of our work's contributions (Reviewers ```wgJu``` and ```hNGa```), presentation (Reviewers ```kTcs``` and ```hNGa```), and **all reviewers** for recognizing our comprehensive experiments.

Below, we summarize the common concerns with our responses and the corresponding modifications.

### More Evaluation
* More causal discovery approaches in [```R2, R4```, Reviewer ```wgJu```; ```R2```, Reviewer ```hNGa```] and [```Appendix D.3.7```].
* Extended  model-based baselines comparison in [```R3```, Reviewer ```hNGa```] and [```Appendix D.3.6```].
* Computational burden analysis in [```R4```, Reviewer ```kTcs```, ```R1```, Reviewer ```hNGa```] and [```Appendix D.3.4```].
* Statiscal analysis in [```R3```, Reviewer ```wgJu```; ```R5``` Reviewer ```kTcs```] and [```Appendix D.3.5```].
* Hyperparameter analysis in [```R5```, Reviewer ```kTcs```] and [```Appendix D.3.3```].

Additionally, we also conducted multi-task generalization evaluation experiments, detailed in ```Appendix D.3.6```.

### Discussion of Limitations and Future Works
* More limitation discussions in [```R2, R4```, Reviewer ```kTcs```] and [```Sec 6```].
* Detailed discussions on future works in [```R5```, Reviewer ```wgJu```] and [```Sec 6```].

We hope that our detailed responses have addressed the reviewer’s concern and remain available for any follow-up questions. Thank you once again for your time and effort!

---

### Meta-Review · Area_Chair_FDKw · 2024-12-17

**Metareview:**

The paper proposes an RL approach called Causal Information Prioritization (CIP), designed to address sample inefficiency by leveraging causal reasoning. The proposed idea combines causal reasoning with empowerment in the context of RL. It is novel and sound. While the idea is creative, the experiments seem confined to specific tasks and assumptions. This does not mean that the experiments are not solid. Most of the concerns are resolved during the rebuttal session however, the scope of the empirical part in this paper is on the narrow side of the spectrum of ICLR papers. Moreover, I encourage the authors to add all the references mentioned in the reviews. Despite what is said, the paper brings new insight to the field. I would accept it for its contribution.

**Additional Comments On Reviewer Discussion:**

The authors have provided more evaluations with different causal discovery approaches and more model-based approaches. More ablations and analysis are done during the rebuttal.

The authors expand the limitation of this work and present clearly about future directions.

The authors addressed most of the concerns.

---

### Decision · Program_Chairs · 2025-01-22

Accept (Poster)